# Inherited Eye Diseases with Retinal Manifestations through the Eyes of Homeobox Genes

**DOI:** 10.3390/ijms21051602

**Published:** 2020-02-26

**Authors:** Yuliya Markitantova, Vladimir Simirskii

**Affiliations:** Koltsov Institute of Developmental Biology, Russian Academy of Sciences, 119334 Moscow, Russia; simir@mail.ru

**Keywords:** retina, inherited retinal diseases, homeobox genes, animal models, molecular genetic diagnostics, gene and cell therapy

## Abstract

Retinal development is under the coordinated control of overlapping networks of signaling pathways and transcription factors. The paper was conceived as a review of the data and ideas that have been formed to date on homeobox genes mutations that lead to the disruption of eye organogenesis and result in inherited eye/retinal diseases. Many of these diseases are part of the same clinical spectrum and have high genetic heterogeneity with already identified associated genes. We summarize the known key regulators of eye development, with a focus on the homeobox genes associated with monogenic eye diseases showing retinal manifestations. Recent advances in the field of genetics and high-throughput next-generation sequencing technologies, including single-cell transcriptome analysis have allowed for deepening of knowledge of the genetic basis of inherited retinal diseases (IRDs), as well as improve their diagnostics. We highlight some promising avenues of research involving molecular-genetic and cell-technology approaches that can be effective for IRDs therapy. The most promising neuroprotective strategies are aimed at mobilizing the endogenous cellular reserve of the retina.

## 1. Introduction

The development of the human eye is controlled by a morphogenetic process that requires precise spatial and temporal gene regulation [1,2]. Perturbation of early eye organogenesis due to genetic factors can result in halting of eye development or multiple eye tissues disorders, and among them degenerations of the retina occupies a special place [3,4,5]. Inherited eye diseases make up a clinically and genetically heterogeneous group of diseases and mutations in which over 260 genes have been proven to be causative. These genes include functionally heterogeneous groups [6]. This review highlights the role of the main homeobox genes associated with inherited eye diseases showing retinal manifestations. Mutations of these genes leading to vision loss in humans have been identified by genetic screenings. Homeobox genes from different classes include retina-specific regulatory genes accepted as critical for eye field specification and retinal cells type differentiation by a broad array of loss- or gain-of-function models. Among these genes are some that are known to cause inherited retinal diseases (IRDs) that disturb the development, function, and survival of rod and cone photoreceptors, ganglion cells, or retinal pigment epithelial cells [4,6,7,8]. In this review, we focused on IRDs associated with single homeobox gene malfunctions as a result of mutations. Mutations in a number of homeobox genes under consideration can manifest themselves the in retina as secondary effects due to impaired functioning of the other eye tissues. It is obvious that an integrated approach should keep in mind the multigenic and systemic nature of a number of retinal/eye diseases to chart the way for appropriate personalized genes and cells therapies technologies and pharmacologic neuroprotection [9,10,11]. We discuss the advantages and disadvantages of modern molecular genetic and cellular approaches, including those that show the most promise for the treatment of a number of retinal neurodegenerative diseases in some of the most striking cases.

## 2. Retinal Organization

The general plan of the retinal architecture is similar across all vertebrates and humans, despite the morphological and functional peculiarities [12]. The retina is composed of two parts: The single-layered retinal pigment epithelium (RPE) on the posterior side and the neuroretina on the anterior side of the eye. The neuroretina is a highly organized multilayered tissue that includes interconnecting layers of specialized cells: Six main types of neurons (photoreceptors, bipolar cells, horizontal cells, amacrine cells, ganglion cells, and interplexiform neurons) and four types of radial glia cells (Muller cells, astrocytes, microglia, and oligodendrocytes) (Figure 1). 

Radial glia of the retina line the bottom and lateral surface of the optic cup, forming the radial layers [14,15]. Three nuclear layers, consisting of different types of sensory neurons, and two plexiform layers (outer and inner) representing synaptic connections between retinal neurons of the border nuclear layer, are distinguished in the retina. The outer nuclear layer (ONL) of the retina includes light-sensitive cells (rods and cones of photoreceptors). The outer segments of photoreceptors are in close interaction with the RPE (single row layer of intensely pigmented epithelial cells). RPE cells are located between the photoreceptors and the choroid and perform a number of physiological functions: Protection of photoreceptors from excess light, transduction of visual signal, retinal homeostasis (growth factor secretion, regulation of ion balance in subretinal space), and phagocytosis of exfoliated discs of outer segments of photoreceptors [16,17]. The RPE is underlain by Bruch’s membrane which consists of the components of the choroid endothelium and the fibrillar layer of the RPE basal plate [18]. The second (inner) nuclear retinal layer (INL) is formed by interneurons: bipolars and amacrine and horizontal cells. Bipolars participate in transmission of visual signal from the photoreceptors into the ganglion, and horizontal and amacrine cells connect all cells of the retina [15]. Ganglion cells are the neurons of the second order, forming the third ganglion nuclear layer. Their axons take part in the formation of the optic nerve [19]. The outer plexiform layer (OPL) is formed by synaptic contacts between rods/cones and bipolar cells, while the inner plexiform layer provides a connection between bipolar and ganglion neurons, as well as a horizontal connection between amacrine and horizontal neurons [20,21]. The INL is where the bodies of Muller glia cells are localized. These retinal macroglia permeates the entire retina from external to internal border basal membranes (formed by outgrowths of these cells) localized on the border of plexiform tissue layers. Muller glia performs structural and neurotrophic functions in the retina [22]. Functions of the other elements of retinal glia, such as astrocytes, are associated with maintaining the structure and metabolic activity of retinal neurons. Astrocytes can secrete vasoactive substances that regulate retinal vascular tone [23]. Retinal microglia present as a resident population of immunocompetent cells [24]. The retina blood supply comes from the central retinal artery feeding the inner retinal layers and choriocapillaries for the outer layers: photoreceptor layer, ONL, and OPL [25]. Retinal neurons, macroglia, and microglia, as well as the wall cells of the microvessels (endotheliocytes and pericytes) interact with each other and form a blood–retinal barrier that regulates the supply of oxygen and trophic factors to retinal neurons and is involved in recycling of metabolic products [26,27,28].

## 3. Homeobox Transcription Factors Expressed in Retina

Retinal development in vertebrate embryogenesis is under strict spatial and temporal regulation by the overlapping gene networks [1,29]. Studies on morphological and molecular characteristics of the eye tissues at different stages of human ontogenesis were conducted many years ago and have contributed to the accumulation of information about various aspects of homeobox genes expression [30,31,32,33,34,35].

Homeobox transcription factors, an evolutionarily conserved class among the transcription factors, are key regulators of developmental processes such as regional specification, cells migration, and differentiation and morphogenesis of the tissues and organs, by regulating the expression of specific sets of target genes. The developing retina is marked by distinct boundaries of homeobox gene expression at different developmental time points (Figure 1). During retinal development the homeobox genes play multiple roles such as regulation of patterning of the retina along the dorsoventral and nasotemporal axes. These genes are essential for the control of proliferation and the choice of cells fate, the order of differentiation of specific neuronal and glial subtypes through instruction signals from surrounding tissues, and retinal cells survival [36,37,38]. Transcription factors expressed in the retina belong to three major groups: basic helix-loop-helix (*bHLH*), forkhead box (*FOX*) and homeobox genes [6,39,40,41]. Among them are the eye field transcription factors (*Rx1*, *Pax6*, *Otx2*, *Optx2*, *Six3*, *Chx10*, *Prox1*, *Dlx1-2*, *Pitx1-2*, etc.) that are activated in the retina at the early stage of embryogenesis and control retinal cell type specialization [42,43,44].

Homeobox transcription factors control the expression of numerous classes of target genes. It is known that, as retinal differentiation proceeds, *Atoh7,* through activation of *Pou4f2* and *Isl1*, leads to the differentiation of ganglion cells; *Foxn4,* via *Neurod1*, *Neurod4*, and *Ptf1a,* provides the fate of amacrine cells; *Rax2* and *Otx2* are responsible for the photoreceptor phenotype, and *Vsx2* and *Ascl1* for the appearance of bipolar cells, but Muller glia arises as a result of overexpression of *Hes1*, *Hes5*, and *Hey2*. A number of reviews considered the interactions between transcription factors to determine the phenotypes of neurons and glia of the retina as the cells are specializing [43]. Homeobox-containing genes continue to be expressed in adult eye tissues, the retina in particular, ensuring eye homeostasis and supporting axon function [4,45]. For example, *PAX6* remains distinctly expressed throughout the lifespan of the human retina, suggesting a role for *PAX6* in the retina after the completion of eye morphogenesis [46]. The role of homeobox genes, such as *Prox1*, as tumor suppressors is well known [47].

The general order of retinal cells differentiation in vertebrates is conserved [48], which makes it possible to model human retinal diseases in experimental animals. Despite the similarities, there are peculiar properties in the retinas of evolutionarily distant species due to the heterochronicity of cellular and molecular processes. These specific features are associated with the length of embryogenesis, adaptation to environmental conditions, the variety of neurons among the general retinal cell types, the expression patterns of key regulatory genes and their isoforms [49,50,51]. These features lead to limitations in modeling human eye/retinal diseases in animal models. Some genes have important differences in temporal expression, such as *CRX* [4,52,53,54]. It is obvious that these differences are associated with species-specific regulatory strategies in genetic programs.

Differences in gene expression levels, including variation in expression depending on age, gender, eye laterality, gene function, and age-by-gender interactions, have been characterized by custom human retinal microarray analysis. These factors contribute significantly to phenotype variation across normal adult retinas. The greater expression variability of the many key genes for retinal function (including photoreceptor-specific genes) may be partially explained by the dynamics of the vision process. Findings show that a significant fraction of gene expression variation in the normal human retina is attributable to identifiable biological factors. Such diversity may result in different levels of disease susceptibility, which is why exploring its sources may provide insights into the pathogenesis of retinal disease [55].

There are homeobox genes specific to the retina (*CRX*, *VAX*, etc.) and others that are widely expressed not only in retina (*PAX*, *SOX*, *OTX2*, etc.), but also in many other organs. Mutations of specific retinal genes are direct and cause primary retinal disorders [52,56,57,58]. The etiologies of many congenital diseases are multigenic in nature. The difficulties in characterizing and diagnosing specific retinal pathologies demonstrate that a number of mutations do not correlate with phenotypic manifestations and many have a pleiotropic effect. Mutations of the genes common to many tissues and organs can often cause the development of secondary eye pathologies (diabetic retinopathy, retinal degeneration in glaucoma, etc.). One well-known example is mutation of the homeobox gene *Prox1*, which causes pathology of the pancreas, and retinal degeneration occurs as a secondary effect [59]. Thus, mutations of different genes can lead to pathological changes in the retina characterized by similar phenotypic traits (age-related macular degeneration (AMD), glaucoma, retinitis pigmentosa (RP), etc). Recent evidence shows that the pathogenesis of anophthalmia and microphthalmia share several molecular factors [35]. Pathologies of other parts of the eye often affect the retina. A more advanced level of high myopia can lead to complications such as glaucoma, cataracts, and retinal detachment. Patients with pathological myopia are at increased risk of retinal detachment due to axial elongation of the globe and peripheral retinal thinning [60].

There has been significant progress in identifying the key genes associated with IRDs and other inherited eye disorders. In the review we mainly paid attention to IRDs associated with single homeobox gene malfunctions as a result of mutations. Disruption of key regulatory genes in the early stages of embryogenesis can result, in the ocular system, in halted eye formation, resulting in various pathological phenotypes of the eye tissues. Distinct clinical manifestations are associated with certain IRDs, but often the diagnosis is complicated by the fact that many genes give rise to more than one disorder [4,6]. In humans, mutations of homeobox genes expressed in the retina have been shown to result in multiple disorders. More than 260 homeobox genes are currently linked to human retinal diseases. This may reflect the essential functions of these homeobox genes throughout embryogenesis or the degree of functional redundancy during retinal development. This is illustrated in Table 1, which includes basic structural properties of reviewed homeobox genes and Table 2, which includes the names all the homeobox genes implicated in the most common IRDs (including some inherited eye diseases with retinal manifestations) and highlights the genes that are shared between different disorders. 

It should be noted that mutations in many genes can be both dominant and recessive. In addition, many of these disorders are part of the same clinical spectrum and have high genetic heterogeneity with both identified and as yet unidentified associated genes [7].

### 3.1. Adnp

The activity-dependent neuroprotector (ADNP) gene family belongs to the zinc finger (ZF) homeobox gene class and includes two genes with significant homologies in human and mouse genomes (*ADNP1/Adnp1*, *ADNP2/Adnp2*). The human *ADNP* gene contains five exons of which the last three are translated. The predicted protein has nine zinc finger motifs, a proline-rich region, a bipartite nuclear localization signal, a partial homeobox domain, a glutaredoxin active site, and a leucine-rich nuclear export sequence [61].

*ADNP* is essential for embryonic brain development [62]. Its expression was demonstrated by RT PCR in adult rat retina [63]. One of the most important cellular processes associated with ADNP is autophagy. Secreted octapeptide NAP, which is derived from ADNP, was found to enhance autophagy, while protecting microtubules [64]. NAP protects retinal ganglion cells against damage induced by retinal ischemia and optic nerve crush [65]. Its neuroprotective actions seem to be mediated by the activation of mitogen-activated protein kinase/extracellular signal-regulated protein kinase (MAPK/ERK) and phosphatidylinositol-3-kinase (PI-3K)/AKT signaling [66].

A set of heterozygous truncating mutations in the *ADNP* gene was identified in patients with Helsmoortel–Van der Aa syndrome (HVDAS). They were characterized by intellectual disability with global developmental delay, gastrointestinal problems, structural brain anomalies, visual problems, and early tooth eruption. *ADNP* is classified as one of three most prevalent autism-causing genes [67]. Ophthalmic findings were remarkable for progressive nystagmus, macular pigment mottling, mild foveal hypoplasia with abnormal macular laminations, persistent rod dysfunction with electronegative waveform, and progressive cone degeneration [68]. HVDAS with ocular anomalies is associated with specific mutations clustering within the “bipartite nuclear localization signal” [69].

*Adnp* knockout in mice resulted in cranial neural tube closure failure and death at the time of neural tube closure (E8.5–9.5) [70]. The *Adnp* haploinsufficient mouse mimics the human ADNP syndrome in terms of synaptic density and gene expression patterns. Daily intranasal treatments with NAP showed significant effects on hippocampal and cerebral cortical expression of the presynaptic *Slc17a7* gene encoding vesicular excitatory glutamate transporter 1 (VGLUT1) [71]. The same effect would be expected for photoreceptor cells that express a splicing variant of VGLUT1 associated with synaptic vesicles [72].

### 3.2. Alx Gene Family

The Alx gene family belongs to the PRD homeobox gene class and comprises three genes in human and mouse genomes (*ALX1/Alx1*, *ALX2/Alx3*, *ALX3/Alx4*). *ALX1* and *ALX3* genes contain four exons each, and encode proteins have a homeodomain and an OAR domain [73].

Vertebrate *Alx* genes are expressed during embryogenesis in forebrain mesenchyme, cranial arches, limb buds, and cartilage [74]. Mouse *Alx* genes show similar developmental expression patterns in the cranial regions of the embryo. From the early stages (E8.5–E9.5), they are markedly expressed in frontonasal head mesenchyme, and later in the first and second pharyngeal arches [74,75]. In the anterior part of the head expression of *Alx1* is first detected in the optic vesicles and restricted to presumptive neural retina in zebrafish embryos. After the completion of retinal differentiation, its expression is restricted to germinal cells at the margin of the retina [76].

In humans, mutations in each of the *ALX* genes have been reported to be associated with congenital craniofacial malformation. There are at least three subtypes of frontonasal dysplasia in humans that are distinguished by their genetic causes (Table 2). 

Mutations of *ALX1* gene lead to frontonasal dysplasia type 3, including eyes that are missing (anophthalmia) or very small (microphthalmia) and low-set ears that are rotated backward [78]. In contrast, mutations of *ALX3* and *ALX4* cause milder forms of frontonasal dysplasia without significant eye manifestations. Individuals with homozygous mutations in the *ALX3* gene have been found to present frontonasal dysplasia type 1, named frontorhiny, characterized by distinctive facial anomalies such as hypertelorism, wide nasal bridge, short nasal ridge, bifid nasal tip, and others [79]. *Alx4* loss-of-function mutations result in frontonasal dysplasia type 2. Heterozygous mice show a single extra digit in the preaxial part of one of the hindlimbs, whereas homozygous mice show a complex phenotype including extensive preaxial polydactyly, tibial anomalies, craniofacial defects, and hypomorphic interfollicular epidermis with reduced suprabasal layers [80,81].

Ectopic expression of *Alx1* in transgenic mice does not disturb development, whereas expression of the form of *Alx1* with deleted OAR domain results in severe cranial and vertebral malformations. It was suggested that the OAR domain of *Alx1* restrains its activity in vivo through its effect on DNA binding. The OAR domain of *Alx3* appeared to have lost most of its function, and its mutations had not visible phenotypic manifestations [82]. *Alx1* homozygous mutant mice are born alive with acrania and meroanencephaly but die soon after birth. This gene is implicated in craniofacial development and is necessary for survival and migration of cranial neural crest cells into the frontonasal primordia [83]. *Alx3*-deficient mice exhibit increased failure of cranial neural tube closure and increased cell death in the craniofacial region [84], but adult mutants do not show any apparent abnormalities, perhaps due the deaths of a number of *Alx3*-null mice during embryogenesis [84]. In contrast, severe nasal clefting and abnormal embryonic apoptosis in *Alx3/Alx4* double mutant mice have been demonstrated [85]. Treatment with *Alx1* antisense oligonucleotides show a reduction of the eye size as well as fewer cell layers in the retinas of zebrafish embryos [76]. Morpholino knock-down of zebrafish *Alx1* expression causes a profound craniofacial phenotype including loss of the facial cartilages and defective ocular development [83]. All of these features correspond with frontonasal dysplasia, a syndrome in humans caused by neural tube closure defects, and suggest a crucial role of *Arl* genes in regulating the migration of cranial neural crest cells into the frontonasal primordia.

### 3.3. CerS2

The ceramide synthase (CerS) gene family belongs to the CERS homeobox gene class and comprises six genes in human and mouse genomes (*CERS1–6/CerS1–6*). They contain seven (*CERS1*) or ten (*CERS2-6*) coding exons and encode transmembrane proteins, each of which synthesizes ceramides with distinct acyl chain lengths [86]. They encode integral membrane proteins of the endoplasmic reticulum and in some cases are associated with mitochondria. All mammalian *CerS* genes contain a unique C-terminal TLC domain, six transmembrane domains, and a Hox-like domain (except *CerS1*), which is derived from DNA-binding homeodomain that lost the first 15 amino acids [87]. Thus, CerS proteins are not DNA-binding transcription factors, but enzymes that participate in lipid metabolism.

Usually more than one type of CerS protein is expressed in a given cell type. *CerS2* transcripts are found at the highest level of all *CerS*, have the broadest tissue distribution and synthesize ceramides containing mainly C20–C26 fatty acids, with little or no synthesis of C16- and C18-ceramides [88].

*CerS2* mRNA is ubiquitously expressed and is highly abundant in many tissues. The highest *CerS2* expression was detected in liver and kidney. In mouse brain, *CerS2* is predominantly expressed in oligodendrocytes and Schwann cells and is up-regulated duringthe period of active myelination. Moreover, *CerS2* genes are located within chromosomal regions that are replicated early within the cell cycle and consist of other features typical of a housekeeping gene, although no other *CerS* genes exhibit these characteristics [86]. Moderate CerS1, CerS2, and CerS4 protein levels were found in the retina. In the cornea, CerS1 and CerS4 are weakly present, whereas CerS2 is strongly expressed. CerS4 is ubiquitously expressed in all retinal neurons (photoreceptors, especially cones, horizontal cells, ON and OFF bipolar cells, GABAergic amacrine cells, ganglion cells) and in Muller glia [89]. In the optic nerve, immunofluorescent reaction on CerS2 was found in the somata of oligodendrocytes, but not along myelinated axons or their myelins’ heath outside of oligodendrocyte somata [90].

Ceramides play a central role in the induction of apoptosis by death receptors and many stress stimuli such as gamma-irradiation and ultraviolet (UV) light [91]. Oxidative stress has also been shown to stimulate an increase in ceramide levels, inducing photoreceptor apoptosis [92].

In humans, rhegmatogenous retinal detachment was found to be associated with a missense coding single-nucleotide polymorphism in the Hox domain of *CERS2*, resulting in an elevated expression level of *CERS2* [93]. A role of *CERS* genes in the progression of AMD was demonstrated, especially in the late stages. In aging, the function of the RPE declines, resulting in an accumulation of degraded photoreceptor outer segments in the form of lipofuscin granules, leading to oxidative stress and retinal inflammation. Oxidative stress-induced Cer biosynthesis genes are involved in photoreceptor cell death [94]. Increased Cer levels in RPE cells raise the levels of inflammatory factors and reactive oxygen species (ROS), which leads to activation of apoptosis [95]. Specific ceramide species were elevated in patients with late-stage AMD. Malondialdehyde–acetaldehyde adducts, oxidation products commonly found in AMD retinas, induced an increase in ceramide levels in retinoblastoma-derived cells in parallel with increased expression of *CerS2* genes [96].

All *CerS*-deficient mice showed reduced electroretinogram amplitudes, with the most severe phenotype in *CerS2* knockout (KO) mice, but had normal retinal morphology. *CerS*-deficient mice showed altered sphingolipid composition in the retina; moreover, retinal C16-sphingomyelins levels were elevated while C18-sphingomyelins levels were reduced [89]. Inhibition of ceramide synthesis by the sphingosine analog FTY720 protects rat retina from light-induced degeneration [97]. In Farber disease, there is accumulation of ceramide in the retina due to a deficiency of ceramidase. The pathological changes are observed in retinal ganglion cells with gross distention and inclusions [98,99]. It is highly possible that *CerS* genes also participate in the pathogenesis of Farber disease. In addition, *CERS2* inactivation leads to myelin sheath defects in axons of the brain and peripheral nervous system [100].

### 3.4. Crx

The *Crx* gene belongs to the PRD class of homeobox genes and is a member of the Otx gene family, comprising three genes in human and mouse genomes (*OTX1/Otx1*, *OTX2/Otx2*, *CRX/Crx*). The human gene consists of four coding exons. The CRX protein includes a homeodomain as well as the WSP motif and the OTX tail [101,102].

In the mammalian retina, Crx is expressed predominantly in postmitotic developing and mature photoreceptors, more weakly in retinal bipolar cells, and in the pinealocytes of the pineal gland [52,103]. In zebrafish *Crx* is expressed in proliferating retinal progenitors and may be involved in patterning the early optic primordium and in promoting the differentiation of retinal progenitors [104].

The Crx homeodomain protein is a transactivator of many photoreceptor/pineal-specific genes in vivo, such as rhodopsin and the cone opsins [105,106]. Crx controls the expression of genes that encode presynaptic proteins associated with the cytomatrix active zone and synaptic vesicles, but not the formation of ribbon synapses, which connect rod and cone photoreceptors to bipolar neurons [107]. CRX also controls outer segment biogenesis and photoreceptor disk renewal [101].

Mutations of the human *CRX* gene are associated with diseases characterized by photoreceptor cell destruction: Autosomal-dominant cone-rod dystrophy 2, Leber’s congenital amaurosis-7 (LCA7) (both autosomal-recessive and -dominant patterns), and, in rare cases, autosomal-dominant RP [108,109,110,111,112]. These diseases are phenotypically and genetically heterogeneous. Cone-rod dystrophy is characterized by primary cone degeneration with significant secondary rod involvement, with additional progressive loss in peripheral vision and night blindness [113]. RP is characterized by progressive loss of rods and cones and causes severe visual dysfunction and eventual blindness in bilateral eyes. In addition to more than 3000 genetic mutations from about 70 genes, a wide genetic overlap with other types of retinal dystrophies has been reported with RP [114,115]. On the cellular level, RP correlates with a predominantly affected rod photoreceptor system. In later stages, the disease may further affect the cone photoreceptors, eventually causing complete blindness. The diseased photoreceptors undergo apoptosis, which is reflected in reduced ONL thickness within the retina, as well as lesions and/or retinal pigment deposits in the fundus [116]. LCA is the most severe childhood retinal dystrophy, characterized by a non detectable electroretinogram and other symptoms soon after birth. LCA is sometimes considered the most severe form of RP [117,118].

The retinas of postnatal day 21 (P21) *Crx*^−/−^ mice were considerably thinner with almost complete absence of outer segments in photoreceptors [107]. In mice homozygous for a targeted null mutation of *Crx*, photoreceptors failed to form outer segments, and eventually degenerated, indicating that *Crx* function is required for their complete differentiation and survival [119]. Outer segment morphogenesis was found to be blocked at the elongation stage, leading to a failure in production of the phototransduction apparatus. *Crx*^−/−^ photoreceptors demonstrated severely abnormal synaptic endings in the OPL [120]. Knockdown of *Crx* by antisense morpholino oligonucleotides resulted in delayed with drawal of RPC from the cell cycle and retardation of retinal differentiation in zebrafish embryos [104]. RNA-seq analysis of three *Crx* mutation knock-in mouse models (*Crx^R90W^*, *Crx^E168d2^*, and *Crx^E168d2neo^*) demonstrated a correlation of graded expression changes in shared gene sets with phenotype severity [121].

### 3.5. Hesx1

The *Hesx* gene family belongs to the PRD homeobox gene class and comprises one gene in human and mouse genomes (*HESX1/Hesx1*). The human gene consists of 4 coding exons encoding a 185-amino acid ORF that is highly conserved compared with the mouse and Xenopus, particularly across the homeodomain sharing 95% and 80% homology, respectively [122]. This gene encodes a conserved homeodomain protein that is required for normal development of the forebrain, eyes, and other anterior structures such as the olfactory placodes and pituitary gland [123]. *Hesx1* is expressed in the rostral part of the chicken neural plate [124] and in the developing forebrain and Rathke’s pouch and is downregulated during pituitary cell differentiation in mice [125]. *Hesx1* plays a role in the control of self-renewal and maintenance of the undifferentiated state in ESCs and mouse embryos [126]. It was demonstrated that it is also required to program hESC neural determination [127]. In normal development its expression is repressed by the *RAX* homeobox gene in presumptive retinal regions of mouse embryos [128].

A homozygous substitution of isoleucine at position 26 by threonine (I26T) in *HESX1* has been associated with anterior pituitary hypoplasia in a human patient, with no forebrain or eye defects, but R160C mutation manifests septo-optic dysplasia characterized by a combination of optic nerve hypoplasia, pituitary hypoplasia, and abnormalities of midline structures in the forebrain [122,129,130].

The absence of *Hesx1* leads to a posterior transformation of the anterior forebrain during mouse development. It is suggested that the mechanism underlying this transformation is the ectopic activation of Wnt/β-catenin signaling [131]. *Hesx1* knockout mice have several defects of midline structures resembling human septo-optic dysplasia. They include a reduction in prospective forebrain tissue, craniofacial and optic nerve dysplasia, and abnormalities of the pituitary gland. Homozygous mutant mice show significant perinatal and postnatal mortality. Most mice surviving past birth display eye defects such as anophthalmia or microphthalmia [123,125].

### 3.6. Hmx1

The *Hmx1* gene belongs to the ANTP class of homeobox genes and is a member of the *HMX/NK5* gene family and comprises three genes in human and mouse genomes (*HMX1/Hmx1*, *HMX2/Hmx2*, *HMX3/Hmx3*). The human gene consists of two coding exons (ENSG00000215612, Ensembl version 98, September 2019). The HMX1 protein includes a homeobox domain [132].

*Hmx1* is expressed in the eye (retina and lens), the craniofacial region, and various nerve ganglia in mice and humans [132,133,134]. In zebrafish embryos, *Hmx1* transcripts were detected mainly in the nasal part of the ganglion cell layer and lens as well as optic vesicles and pharyngeal arches by 24–32 hpf. Before this stage, transcripts were more uniformly expressed in the optic vesicle [135].

A loss-of-function mutation in *HMX1* causes an oculoauricular syndrome in humans. This syndrome is characterized by microphthalmia, microcornea, cataract, ocular coloboma, retinal pigment abnormalities, rod-cone dystrophy and anomalies of the external ear [132,136,137]. Homozygous missense mutation within the homeodomain of *HMX1* was demonstrated to be associated with uveoretinal colobomas caused by a failure of ectodermal optic vesicle fissure closure [138].

A single C > T mutation that changes Glu65 to an amber stop codon in the *Hmx1* gene displays microphthalmia, protruding ear and craniofacial malformations in mice [139]. Delayed withdrawal of retinal progenitors from the cell cycle resulting in retarded retinal differentiation and microphthalmia were observed after morpholino-mediated knockdown of *Hmx1* in zebrafish [135].

### 3.7. Lmx1B

The *Lmx* gene family belongs to the LIM homeobox gene class and comprises two genes in human and mouse genomes (*LMX1A/Lmx1a*, *LMX1B/Lmx1b*). The human *LMX1B* gene contains eight exons and encodes protein featuring two LIM domains in their amino termini and a centrally located homeodomain, and a putative transcriptional activation domain at the COOH terminus. The LIM domain contains two tandemly repeated, cysteine-rich, double-zinc finger motifs that can be recognized by a number of co-factors [140,141].

In the mouse eye, *Lmx1b* is expressed in periocular mesenchyme and regulates anterior segment (cornea, iris, ciliary body, trabecular meshwork, and lens) development; mice lacking functional *Lmx1b* exhibit numerous abnormalities, including a lack of ciliary bodies and iris stroma, and corneal dysplasia [142].

Heterozygous mutations in the LIM homeodomain of *LMX1B* resulted in nail–patella syndrome (NPS), which is characterized by developmental defects of dorsal limb structures and nephropathy. Up to 50% of NPS patients exhibit clinical signs of primary open-angle glaucoma [143,144,145]. It was proposed that primary structural changes in the membranes of the lamina cribrosa and the trabecular meshwork may play a role in glaucomatous optic nerve damage in NPS and may result from disrupted collagen expression and/or an irregular arrangement or deposition of collagen fibrils [143].

It was strongly confirmed that haploinsufficiency is the principal pathogenetic mechanism of NPS in humans. A difference in dosage sensitivity for the *LMX1B/Lmx1b* gene between humans and mice was suggested [146].

An N-ethyl-N-nitrosourea-induced missense substitution, V265D, in the homeodomain of mouse *Lmx1b* abolishes DNA binding and causes glaucomatous eye defects in heterozygotes. A profound reduction in the number of retinal ganglion cells and optic nerve cupping was observed in some cases [147].

Anti-sense morpholinos against *Lmx1b.1* and *Lmx1b.2* isoforms resulted in defective migration of periocular mesenchymal cells around the eye and led to apoptosis of these cells. These defects in the periocular mesenchyme are correlated with a failure of fusion of the choroid fissure or, in some instances, more severe ventral optic cup morphogenesis phenotypes. The retinas of *Lmx1b* morphants showed defective nasotemporal patterning and lamination was delayed [148].

### 3.8. Meis1

The myeloid ecotropic insertion site 1 (*Meis1*) gene belongs to the three amino acid loop extension (TALE) class of homeobox genes and is a member of the *Meis* gene subfamily, comprising three genes in human and mouse genomes (*MEIS1/Meis1*, *MEIS2/Meis2*, and *MEIS3/Meis3*). The *MEIS1* gene consists of 13 coding exons [149] (ENSG00000143995, Ensembl version 98, September 2019) and encodes protein containing a DNA-binding homeodomain toward the carboxy terminus and two protein–protein interaction domains toward the amino terminus (MEIS-A and MEIS-B). TALE homeodomain differs from the classical homeodomain by the insertion of three additional amino acids [150,151].

*Meis1* is required to correctly specify both dorsoventral and nasotemporal identity in the zebrafish retina. It is initially expressed throughout the eye primordium. Later, *Meis1* becomes repressed as neurogenesis is initiated, and its expression is confined to the ciliary margin, where the retinal stem population resides. Thus, *Meis1* maintains RPC in a rapidly proliferating state. Cell cycle control is mediated through regulation of c-myc and/or cyclinD1 [152,153,154].

Morpholino knockdown of *Meis1* causes a delay in the G1-to-S phase transition of retinal cells. Consequently, fish, chicks, or mice with compromised *Meis1* function are microphthalmic [152,153,155]. Knockdown of *Meis1* expression diminishes endogenous *Foxn4* expression and affects development of horizontal and amacrine neurons in mice [156]. By combining an analysis of *Meis1* loss of function and conditional *Meis1* functional rescue with ChIP-seq and RNA-seq approaches, it was shown that in mice *Meis1* coordinates, in a dose-dependent manner, retinal proliferation and differentiation by regulating genes responsible for human microphthalmia and components of the Notch signaling pathway [157].

Studies in animal models of the orthologous genes identified overlapping phenotypes for most factors, confirming the conservation of their function in vertebrate development. However, despite whole exome/genome testing more than half of patients currently remain without a molecular diagnosis. Anophthalmia and microphthalmia are part of the same clinical spectrum and have high genetic heterogeneity, with >90 identified associated genes. Currently, mutations in known genes explain less than 40% of these conditions therefore additional causative factors need to be discovered [35]. The *MEIS1* gene is the best gap-filling candidate for decoding genetic pathways associated with anophthalmia and microphthalmia.

### 3.9. Msx2

The human *MSX* gene family consists of two members (*MSX1*, *MSX2*), but the mouse has an additional *Msx3* gene. These genes belong to the ANTP class of homeobox genes. *MSX2* consists of two coding exons and encodes proteins with homeodomain. These regulatory proteins function as transcriptional repressors and are widely expressed in many organs, particularly at the sites where epithelial–mesenchymal interactions take place [158,159]. In chick embryos, *Msx2* was widely distributed in the dorsal neural retina and pigment epithelium as well as in the retro-ocular mesenchyme [160]. Mouse *Msx3* is only expressed in the dorsal neural tube [161].

*Msx2* drives osteogenic differentiation of multipotent mesenchymal progenitors and yet suppresses adipogenesis [162]. Direct interaction with DNA is not required for Msx2 suppressor function. Msx2 suppresses transcription via protein–protein interactions with components of the basal transcriptional machinery [163]. Msx2 controls diverse processes during brain development (e.g., apoptosis, neuronal specification and differentiation) mediating distinct stage-dependent roles of BMPs in dorsal neural-tube patterning [159].

A gain-of-function mutation in human *MSX2* causes autosomal-dominant Boston-type craniosynostosis, characterized by overgrowth of skull bones [164,165], while loss-of-function mutations result in *MSX2* haploinsufficiency and lead to reduced ossification of the parietal bones, enlarged parietal foramina and aberrant closure of the sagittal suture [166]. The case of a child with bicoronal synostosis and cutaneous syndactyly, who presented iridal and chorioretinal colobomas due to *MSX2* gene duplication has been recently reported [167].

In mice, overexpression, misexpression, or deficiency of *Msx2* impedes osteoblast differentiation and leads to many craniofacial deformities. *Msx2* null mutation causes pleiotropic defects in bone growth and in the ectodermal organs, including the teeth, hair, and mammary glands, and impaired chondrogenesis [168]. These mice display defective skull ossification and marked reduction in bone formation associated with decreased osteoblast numbers, which is very similar to what is observed in humans. It has been shown that *Msx2* promotes osteoblast differentiation independently of *Runx2* and negatively regulates adipocyte differentiation through inhibition of PPAR and the C/EBP family [169]. Transgenic mice with a dominant gain-of-function mutation (Pro7His) of the *Msx2* gene have been constructed. These mice exhibited precocious fusion of cranial bones and craniosynostosis and provided an excellent model of human craniosynostosis [170]. Overexpression of the wild-type form of the *Msx2* gene in Xenopus resulted in embryos with a loss of anterior structures, including head and eyes [171].

*Msx2* appears to play dual roles in promoting apoptosis and determining retinal fate. Overexpression of mouse *Msx2* led to microphthalmia and optic nerve aplasia, suppressing *Bmp4* expression in the optic vesicle before lens induction, whereas *Bmp7* expression was upregulated. Retinal apoptosis together with an overall reduction in proliferation resulted in thinning of the retina and microphthalmia [172]. Antisense disruption of mouse *Msx2* during early stages of neurulation produced hypoplasia of the maxillary, mandibular, and frontonasal prominences and somite, and neural tube, eye, and somite abnormalities. Eye defects consisted of enlarged optic vesicles, which may ultimately result in microphthalmia. Histological analysis revealed thinning of the neuroepithelium in the diencephalon and optic vesicle and hypoplasia of the mid- and forebrain regions [173]. Overexpression of *Msx2* delayed the expression of RGC-specific differentiation markers (Math5 and Brn3b), which showed that it could affect the timing of RGC fate commitment and differentiation by delaying the timing of cell cycle exit of retinal progenitors [174]. *Msx2* can suppress the differentiated state of RPE cells and promote their differentiation into neural cell types. *Msx2*-transfected cultures of chick RPE contained fewer cells expressing the RPE marker, Mitf, and more cells expressing class III b-tubulin, a neuronal marker. In addition, a small proportion of *Msx2*-transfected cells acquired a neural-like morphology [175].

### 3.10. Otx2

The *Otx2* gene belongs to the PRD class of homeobox genes. This gene family includes three genes in human and mouse genomes (*OTX1/Otx1*, *OTX2/Otx2*, and *CRX/Crx*). The *OTX2* gene consists of five exons (ENSG00000165588, Ensembl version 98, September 2019) and encodes protein with homeodomain [176]. The OTX2 protein contains a homeodomain responsible for DNA binding, SGQFTP and SIWSPA motifs involved in protein–protein interactions, and two C-terminal tandem OTX-tail motifs responsible for transactivation [177].

The *Otx2* gene plays a critical role in forebrain and eye development. At later embryonic stages, the gene is expressed in several regions of the brain and in sensory organs such as the inner ear, retina, and olfactory epithelium. In the embryonic mouse, Otx2 was found in RPE cells and a restricted subset of retinal neurons, including ganglion cells. In the postnatal and adult eye, it was detected in the nuclei of RPE and bipolar cells, and restricted to a small volume at the inner periphery of nuclei of photoreceptors [178,179]. In the differentiating outer retina, Otx2 is expressed in progenitors of both photoreceptor and bipolar cells. Beyond developmental stages, Otx2 expression is abundantly maintained in RPE, photoreceptors, and bipolar cells [179].

A majority of the known *OTX2* mutations involve homeobox and the last exon of the gene [177]. Heterozygous mutations of *OTX2* cause severe ocular malformations, which range from bilateral anophthalmia to retinal defects resembling LCA and pigmentary retinopathy [180]. *OTX2* loss-of-function mutations are frequently associated with coloboma, optic nerve hypoplasia or aplasia, microcephaly, brain, and pituitary anomalies and combined pituitary hormone deficiency [181,182,183]. Heterozygous mutations in *OTX2* associated with early-onset retinal dystrophy with atypical maculopathy and bilateral microphthalmos have been reported [184]. Most *OTX2* mutations reported to date originate from premature truncation of the protein product. As a rule, pituitary anomalies seem to be more strongly associated with mutations that occur in the second half of *OTX2*, after the homeodomain and SGQFTP motif [185].

Complete elimination of *Otx2* in knockout mice results in the absence of the forebrain and embryonic fatality due to the absence of the rostral neuroectoderm fated to become forebrain, midbrain, and rostral hindbrain [186]. *Otx2* hemizygotes survive until birth and demonstrate anomalies in RPE and retina development. Depending on the genetic background, *Otx2*^+/−^ embryos show variable phenotype (acephaly, holoprosencephaly, short nose, anophthalmia/microphthalmia, agnathia/micrognathia, or normal phenotype) [187]. *Otx2* conditional knockout mice exhibited a total absence of rods and cones in the retina due to their cell fate conversion to amacrine-like cells [188]. A number of mature bipolar cells were diminished in postnatal mice with bipolar cell-specific *Otx2* conditional-knockout [189]. In the adult retina, loss of Otx2 protein after induced conditional knockout caused slow degeneration of photoreceptor cells due to dramatic changes of RPE consisting of reduction in melanin content, extensive vacuolization, and loss of RPE contact with disc-containing photoreceptor outer segments [190].

So, the gene is required for RPE specification and serves as a key regulator of photoreceptor genesis and differentiation, and is required after birth for bipolar cell terminal maturation and long-term maintenance of photoreceptors [177].

### 3.11. Pax2

The *Pax2* gene belongs to the PRD homeobox gene class and is a member of the *PAX258* gene subfamily (Pax group II). It comprises three genes in human and mouse genomes (*PAX2/Pax2*, *PAX5/Pax5*, and *PAX8/Pax8*). *PAX2* contains 12 exons and encodes protein with a paired domain, an octapeptide domain, a partial homeodomain (which still has DNA-binding properties), and the transactivation domain. None of the known transcripts contain all 12 coding exons of the *PAX2* gene lacking either exon 6 or exon 10 [191,192].

During human and mouse development, *PAX2/Pax2* is expressed in the developing eye, ear, kidney, midbrain, hindbrain, and spinal cord. In the mouse eye, *Pax2* expression begins in the optic vesicle and becomes restricted to the ventral half of the optic cup and stalk and later to the optic disc and nerve. After closure of the retinal fissure, Pax2 protein is lost from the ventral retina; however, it is localized mostly to the proximal regions, destined to contribute to the optic nerve and optic chiasm [193,194]. Pax2-expressing cells in the developing rat and human optic nerve are exclusively astrocyte precursors and astrocytes controlling oligodendrocyte differentiation in the optic stalk. Coloboma formation may result from impaired astrocyte differentiation during development [195,196]. In rodents, cells of astrocytic lineage migrate into the retina through the optic nerve head as a mixture of precursor cells and immature perinatal astrocytes, and then spread across the nerve fiber layer toward peripheral margins of the retina. The migration of astrocytes into the retina is followed by the invasion of endothelial cells to form the retinal vasculature, which in turn promotes astrocyte differentiation [197]. In the adult human retina, mature perinatal astrocytes were restricted to a region surrounding the optic nerve head, whereas adult astrocytes were apparent throughout the vascularized retina. A cluster of Pax2 cells was also present in a small region surrounding the optic nerve head at the ventricular surface of the developing retina, which suggests the existence of two distinct sites of astrocytic differentiation [195]. Pax2 is rapidly downregulated in explanted optic nerves that generate neurons, and its overexpression by electroporation in the optic nerve, or ectopically in the neural tube, is sufficient to block neuronal differentiation and allow glial development. It has been suggested that *Pax2* is able to regulate a switch between neuronal and glial fates inhibiting neurogenesis or inducing gliogenesis in the optic nerve [198]. In chick and zebrafish embryos, Pax2 is expressed also by Muller glia cells in the central retina, but similar expression could not be detected in mouse, rat, guinea pig, rabbit, or pig retina [199,200]. Pax2 is expressed in astrocyte precursor cells and mature astrocytes of the optic stalk/nerve and glial cells of a vascular structure in avian and reptile species called the pecten [196,201].

The majority of mutations in *PAX2* are pathogenic or likely pathogenic and expected to result in significant truncation of the PAX2 protein through a shift in the reading frame or the introduction of a premature stop codon. These mutations were located in all four known functional domains of this gene, but more often in exons 2–4 (encoding the paired domain) and exons 7–9 (encoding the transactivation domain) [192,202]. *PAX2*-related disorder was originally termed renal coloboma syndrome (also known as papillorenal syndrome) and characterized by renal (hypodysplastic kidneys) and optic disc anomalies. Abnormal renal structure or function is noted in 92% and ophthalmologic abnormalities in 77% of affected individuals [203]. Optic nerve malformations include optic nerve coloboma, optic nerve dysplasia, morning glory anomaly, and cystic malformation of the optic nerve. Other eye malformations may include retinal coloboma, microphthalmia, and macular dysplasia. Less common associated eye malformations include abnormal RPE, abnormal retinal vessels, and chorioretinal degeneration [204,205].

The developmental mechanism underlying the optic nerve abnormalities observed in *PAX2*-related disorder is under investigation in animal models; in both mouse and zebrafish, homozygous loss-of-function alleles of *Pax2/Pax2a* resulted in failed optic fissure closure [206]. Several mouse models of renal coloboma syndrome have been described. After deletion of exon 1 and 2 of the *Pax2* gene, the optic tracts remained totally ipsilateral due to a genesis of the optic chiasma. These mutants showed extension of the pigmented retina into the optic stalks and failure of the optic fissure to close, resulting in coloboma [207]. A frame shift mutation (*Pax2^1Neu^*) with a 1-bp insertion in the *Pax2* gene has been described. The same mutation was in a human family with renal–coloboma syndrome. Heterozygous mutant mice exhibited defects in the kidney, optic nerve, and retinal layer of the eye. In homozygous mutant embryos, development of the optic nerve, kidney, and ventral regions of the inner ear was severely affected, as in the case of the same mutation in humans [208]. Mice heterozygous for a *Pax2* missense mutation paired domain showed reduced optic nerve axon numbers [209]. The hypomorphic mutation Opdc for *Pax2* has been reported. In homozygotes, the phenotypic consequence of this mutation on the development of the eye and ear was similar to that reported for null alleles of *Pax2*, but homozygotes had undisturbed kidney development under some strain backgrounds [210]. Nearly all homozygous mutant mice are anephric, with most not surviving past the immediate postnatal period. These mice have midbrain/hindbrain malformations, cranial neural tube closure defects (exencephaly), and absent cochlea. The ocular phenotype is characterized by a failure of closure of the optic fissure and failed basement membrane dissolution. *Pax2* plays a critical role in optic chiasm development and in the guidance of axons along the optic tract. Null mutant *Pax2* mice develop an abnormal chiasm in which all optic axons are prevented from projecting across the midline into the contralateral optic tract [207].

### 3.12. Pax6

The *Pax6* gene belongs to the PRD homeobox gene class and is a member of the *PAX46* gene subfamily (Pax group IV). It comprises two genes in human and mouse genomes (*PAX4/Pax4*, *PAX6/Pax6*). *PAX6* contains 14 exons and encodes protein with two DNA binding domains, a paired domain and a homeodomain, and one proline/serine/threonine-rich transactivation domain [211]. The paired domain is a bipartite DNA binding structure composed of two helix–turn–helix motifs (the N-terminal PAI and C-terminal RED subdomains) that are separated by a flexible linker. These subdomains have tissue-specific effects on development and gene expression [212].

In vertebrates, this factor is essential for normal development of several organs, including the brain, pancreas, and eye [213]. *Pax6* is expressed in the anterior neural plate in the cells that will give rise to the optic vesicle. *Pax6* activity is required for the establishment and maintenance of dorsal and nasotemporal characteristics in the optic vesicle and, later, the optic cup [214]. Although *Pax6* is not required for optic vesicle formation, it does play a role in subsequent steps of retinogenesis. At the optic cup stage, *Pax6* seems to be required for cell proliferation and differentiation. Following optic cup formation, *Pax6* is downregulated in the optic stalk and the RPE, but retained in the neuroretina. *Pax6* plays a role in determining neuronal cell fate in the retina and, together with *Pax2*, directs the determination of RPE controlling the expression of the melanocyte determinant Mitf [215]. In differentiating RPE, Pax6 is required for promoting its melanogenic program [216]. Pax6 maintains the multipotency and proliferation of RPCs through the activation of proneural genes [217]. Expression in the retina is maintained in proliferating RPC, while it is downregulated in most cells upon differentiation. After evagination of the optic vesicle, Pax6 becomes restricted to all proliferating cells of the pigment epithelial and neural layers of the retina [218]. Pax6 is diffusely expressed in the undifferentiated retinal neuroepithelium as cells become postmitotic and arranged in a characteristic laminar pattern; however, *Pax6* remains strongly expressed in putative ganglion, amacrine, and horizontal cells, while becoming conspicuously downregulated in photoreceptor progenitors and in the cells occupying the Muller/bipolar region of the INL [219,220]. *Pax6* has been assumed to play a several roles during early retinal development: Proliferation of retinal cell progenitors [221], maintenance of multipotent progenitor potential [222], and regulation of timing of differentiation and cell fate determination [223]. In the adult human retina, PAX6 proteins are expressed in the ganglion cells and INL [46]. Pax6 has been detected immunochemically in amacrine and ganglion cells of adult retinas of rats and goldfish [224,225]. In the adult mouse, *Pax6* was found to be essential for generating late-born glycinergic amacrine cells, along with most bipolar cell subtypes. Overexpression of Pax6 greatly increased the non-GABAergic/non-glycinergic amacrine cells, suppressed generation of both glycinergic amacrine cells and bipolar interneurons, and disrupted rod photoreceptor morphogenesis [226]. Pax6 has been proposed to regulate the maintenance of horizontal cells through the activation of ONECUT1 and ONECUT2 transcription factors [227]. It was demonstrated that development of the lens from the surface ectoderm requires a higher gene dose of *Pax6* than development of the retina from the optic vesicle [228].

More than 500 different mutations have been described that affect *PAX6* and its regulatory regions. The majority of *PAX6* mutations result in null alleles and consequent *PAX6* haploinsufficiency and are known to result in aniridia, an autosomal-dominant disorder that is marked by the complete or partial absence of the iris, often combined with cataracts, glaucoma, nystagmus, and foveal and optic nerve hypoplasia. Missense mutations reported in ~12% can potentially cause partial loss-of-function or gain-of-function generating single amino acid substitutions, and lead to less severe ocular abnormalities (foveal hypoplasia, Peters anomaly, congenital cataracts, microphthalmia, optic nerve hypoplasia, coloboma, and anterior segment dysgenesis) [229,230]. Patients with a *PAX6* mutation occurring at the donor splice site of intron 4 have been reported to have congenital nystagmus with anterior segment anomalies (mainly iris hypoplasia or coloboma) associated with foveal hypoplasia [231]. Microphthalmia/anophthalmia/coloboma phenotypes are significantly associated with recurrent heterozygous *PAX6* missense mutations in the paired domain that are likely to disrupt the PAX6–DNA interaction [232,233]. Anophthalmia is associated with homozygous *PAX6* variants, where there is biallelic loss of function. The affected individuals are usually still born or die soon after birth with severe brain abnormalities [234,235]. A heterozygous, likely pathogenic, variant in PAX6 associated with microphthalmia was reported to have a highly conserved valine residue in the homeodomain [236].

Homozygous *Pax6*^−/−^ mice, with two nonfunctional alleles, die at birth with no eyes or nose and with brain abnormalities [237,238]. Heterozygous *Pax6*^+/−^ mice are viable and fertile but have a range of eye abnormalities, such as microphthalmia, iris hypoplasia, cataracts, a thin corneal epithelium with fewer cell layers, corneal opacity, failure of the lens to separate completely from the corneal epithelium, retinal dysplasia, coloboma, and adhesions between the lens and cornea or between the iris and cornea [239]. Embryos with putative null *Pax6* mutations also show severe eye abnormalities and changes in brain development resembling human aniridia phenotypes [240]. In mutant mice with Pax6 activity ranging between 100% and 0%, the extent of eye development was progressively reduced as Pax6 activity decreased, to very early termination of eye development at the lowest levels of Pax6. Development of the lens and cornea is more sensitive to reduced levels of *Pax6* activity than development of the retina [241]. Overexpression of *Pax6* in *PAX77*^+/+^ mice prevents normal development of the retina from about E14.5 and results in microphthalmia, retinal dysplasia, and defective retinal ganglion cell axon guidance in postnatal life [242]. The *Pax6*-deficient mouse model of aniridia (*Pax6^Sey^*^+/−^) has been used for topical application of nonsense mutation suppression drugs on adult eyes. Nonsense suppression not only inhibited disease progression but also stably reversed corneal, lens, and retinal malformation defects and restored electrical and behavioral responses of the retina [243].

### 3.13. Rax

The *Rax* gene family belongs to the PRD class of homeobox genes and includes two genes in the human genome (*RAX*, *RAX2*) and only one gene in the mouse genome (*Rax*). *RAX* contains three exons and encodes a protein with an octapeptide, a DNA-binding paired-type homeodomain, an Rx domain, and an OAR domain [244,245]. The octapeptide functions in transcriptional repression through interaction with Groucho family corepressors [246].

*Rax* is initially expressed in the anterior neural region of developing mouse embryos, and later in the retina, pituitary gland, hypothalamus, and pineal gland [247]. In the early mouse embryo, *Rax* is expressed in the anterior neural fold, including areas that will give rise to the ventral forebrain and optic vesicles. *Rax*, mainly through activation of transcriptional repressors TLE2 and Hes4, is necessary and sufficient to inhibit endomesodermal gene expression in retinal precursors of the eye field [248]. *Rax* is expressed in all RPC in the neuroepithelium of the developing retina and in RPC in the ciliary margin of the mature retina in mice and frogs [249,250,251]. Its expression is progressively downregulated as neuronal differentiation proceeds, except for photoreceptors and Muller glia cells [249,250]. It has been demonstrated that Rax proteins are necessary for normal photoreceptor development and expression of normal levels of markers of differentiated photoreceptors in zebrafish and Xenopus [250,252]. Rax expression is retained in the ciliary marginal zone, which is a source of retinal stem cells in adult fish and amphibians [252].

*RAX* mutations in humans have been reported, including anophthalmia, microphthalmia, and eye coloboma [253,254], in some cases associated with sclerocornea [57] or severe cerebral malformation [255].

*Rax*-null homozygotes exhibit perinatal mortality, anophthalmia, and anterior nerve and craniofacial defects [256]. *Rax*^−/−^ mouse embryos have no visible eye structures, while heterozygotes for the *Rax* mutation are apparently normal [257]. In zebrafish, knockdown of *Rax* over the period of eye organogenesis resulted in severely reduced proliferation of retinal progenitors, the loss of expression of transcription factor Pax6, delayed retinal neurogenesis, and extensive retinal cell death [258]. Knockdown of *Rax* using translation-inhibitory antisense morpholino oligonucleotides also resulted in anophthalmia [250,257]. Overexpression of *Rax* led to the loss of forebrain tissue and the ectopic formation of retinal tissue in zebrafish and Xenopus [257,259]. Moreover, for the same induction, expression of the Rax homeodomain alone is sufficient [260]. In summary, gain- and loss-of-function studies of *Rax* in different species have shown this gene to be necessary for the formation of anterior brain structures and eyes, and sufficient to induce retinal and neural tube hypertrophy [257,261,262].

### 3.14. Rax2

The *Rax2* gene belongs to the PRD class of homeobox genes. *RAX2* contains three exons and encodes a protein with a DNA-binding paired-type homeodomain, an Rx domain, and an OAR domain. Rax2 proteins of all tetrapods are shorter than Rax1 proteins and lack the octapeptide motif. There is no *Rax2* gene in mouse and other rodent genomes [245,263]. Two *Rax* genes have been identified in chicks (*cRax* and *cRaxL/cRax2*) [264] and in Xenopus [265].

*Rax2* are among the earliest markers of the eye field, being initially expressed in the anterior neural region of head-fold-stage embryos, but later becoming restricted to the neural retina and ventral hypothalamus [266]. Human *RAX2* is expressed in the ONL and INL of the adult human retina and act synergistically with CRX and NRL to modulate the expression of photoreceptor genes [263]. RAX2 is one of the top expressed transcription factors in the adult human retina, which suggests a major role in the regulation of retinal transcription [267]. The chick *RaxL/Rax2* gene is expressed in both RPC and early-developing photoreceptors. It is not sufficient to promote photoreceptor cell fate choice, but is required for cone photoreceptor cell differentiation [264]. In zebrafish embryos, the level of expression of *Rax2/Zrx2* is greater than *Rax/Zrx1* in immature photoreceptors at comparable stages, whereas expression of the latter is enhanced during maturation of photoreceptors. Expression of both genes continues in the adult retina, exclusively in cone, but not rod, photoreceptors [268].

A heterozygous mutation in *RAX2* inherited in an autosomal-dominant fashion has been reported to be associated with mixed cone and rod dysfunction and AMD [263,269]. Biallelic *RAX2* sequence and structural variants were found in patients with nonsyndromic autosomal-recessive RP [270].

The absence of *Rax2* in mouse has made functional analysis and modeling of the associated human diseases more difficult. The expression of a putative dominant negative allele of a chick *Rax2* gene caused a significant reduction in the level of expression of cone photoreceptor genes [264]. Morpholino-based knockdown of *Rax2/Rx-L* in Xenopus appeared to impair late retinogenesis and reduce photoreceptor-specific gene expression [271]. Animal models demonstrate that *Rax2* is required for cell proliferation and differentiation within the retina by regulating the spatial expression of photoreceptor-specific genes in late retinogenesis.

### 3.15. Vax1

The *Vax1* gene belongs to the ANTP class of homeobox genes. This gene family includes two genes in human and mouse genomes (*VAX1/Vax1*, *VAX2/Vax2*). *VAX1* contains four exons and encodes a protein with DNA-binding paired-type homeodomain [272]. Vax1 protein can be secreted and penetrate into axons of RGC and other neurons [273].

In vivo, expression of *Vax1* is highly restricted to ventral anterior regions of the developing CNS, including the glial precursor cells of the optic stalk and their descendent astrocytes of the optic nerve, and regions containing the glia that are thought to guide the formation of midline commissural tracts. At E18.5, *Vax1* expression is visible in the ventral optic stalk, the glial cells of the optic nerve, the optic chiasm, and the rostral diencephalon [272,274,275]. *Vax1* acts in concert with *Vax2* to ventralize the developing eye field of mouse embryos. They allow for development of the optic nerve by inhibiting development of the retina through repression of the *Pax6* and *Rax* genes and therefore are involved in the partitioning of the developing visual system in the eye and optic nerve [275,276]. In *Danio rerio*, *Vax1* is expressed in the ventral portion of the developing eye [277]. It was discovered that Vax1 is secreted from ventral hypothalamic cells and diffuses to RGC axons, where it promotes axonal growth [273]. Microphthalmia associated with cleft lip and palate and agenesis of the corpus callosum caused by homozygous mutation in the *Vax1* gene was reported [278].

*VAX1* has been also identified as a candidate gene playing a role in human nonsyndromic cleft lip with or without cleft palate, which can be accompanied by brain anomalies, eye coloboma, and syndromic microphthalmia [279,280].

*Vax1*-null mice undergo neonatal mortality due to severe holoprosencephaly and cleft palate. The eyes of mutant homozygotes display a failure of the optic disks to close, leading to coloboma and loss of the eye–nerve boundary. Retinal axons fail to penetrate the brain and form an optic chiasm. The periphery of nearly the entire length of the *Vax1* mutant nerve was frequently pigmented by cells of the RPE [272,275,276]. In *Danio rerio*, injection of antisense morpholinos against *Vax1* resulted in colobomas and reduced retinal pigment at the site of the choroid fissure [277]. In chicks, ectopic expression of *cVax* and *mVax2* led to ventralization of the early retina. Moreover, the projections of dorsal but not ventral ganglion cell axons onto the optic tectum showed profound targeting errors [281]. Overexpression of Xenopus *Vax1/Xvax1* after injection of its mRNA into one or two blastomeres at the two-cell stage primarily affected eye development in a dose-dependent manner. Low doses resulted in only a slight reduction in eye diameter, whereas increasing doses produced cyclopic and microcephalic embryos and could inhibit head and eye formation completely [275].

### 3.16. Vax2

The *Vax2* gene belongs to the ANTP class of homeobox genes. This gene family includes two genes in human and mouse genomes (*VAX1/Vax1*, *VAX2/Vax2*). *VAX2* contains three exons and encodes a protein with DNA-binding paired-type homeodomain [282].

In mice, *Vax2* transcripts are detected by whole-mount in situ hybridization almost exclusively in the ventral half of the optic vesicle from E9.0 onward. At E12.5, *Vax2* is highly and almost exclusively expressed in the prospective inferior neural retina. Labeling seems to be equally strong in the deep and superficial layers of the differentiating neural retina. Expression is also detected at a lower level from the inferior neural retina along the entire optic nerve and stalk. Thus, at later stages the expression domains of *Vax1* and *Vax2* become segregated, with *Vax1* predominantly found in the ventral optic stalk and *Vax2* in the ventral retina [282,283]. *Vax2* inactivation leads to altered expression of genes metabolizing retinoic acid and altered asymmetric expression of cone photoreceptor genes (*S-Opsin* and *M-Opsin*). Moreover, adult *Vax2* mutant mice revealed disorganization of the nerve fiber layer within the retina. Therefore, *Vax2* is confirmed to not only play a crucial role in the development of the embryonic ventral retina but also continue to function in the mature retina, and may play a role in the generation of the visual streak or the macula [284]. Vax2 was found to shuttle between the nucleus and cytoplasm in retinal differentiation. This shuttling is controlled by phosphorylation. Disruption of phosphorylation and constant nuclear localization of Vax2 protein in the chick optic vesicle results in constitutive repression of *Pax6*, and leads to the formation of an eyeless embryo [285].

It has been reported that *VAX2* in humans has two isoforms; both forms are enriched in neuronal tissues, including the retina. In monkey retina, *Vax2* was observed to localize either to the nucleus (ganglion cells) or to the cytoplasmic compartments (outer segment of cone photoreceptors) depending on the retinal cell type [286]. The *VAX2* p.Leu139Arg variant, harbored by a cone dystrophy patient with loss of outer retinal tissue at the fovea, was identified. Mutant protein was mislocalized and degraded, forming aggresomes [287]. Bilateral rod/cone photoreceptor dystrophy and mild optic atrophy in the patient, with complete deletion of *ATP6V1B1* and disruption of the *VAX2* open reading frame, was revealed. Similar changes were not detected in an adult harboring a disruptive mutation in *ATP6V1B1* usually associated with distal renal tubular acidosis [288].

Colobomas were rare and milder in *Vax2* homozygous null mutants, but *Vax1* and *Vax2* double-mutant mice had severe colobomas that were fully penetrant [289,290]. The involvement of this gene in the development of ocular coloboma in humans was predicted [289]. Injections of antisense morpholinos against *Vax2* or *Vax1* resulted in colobomas and reduced retinal pigment at the site of the choroid fissure in *Danio rerio* [277]. *Vax2* overexpression in mouse, frog and chick embryos ventralizes the eye. Furthermore, *Vax2* overexpression induces a striking expansion of the optic stalk, a structure deriving from the most ventral region of the eye vesicle [282,283,289]. This ventralization is mediated, at least in part, by Vax2 and Vax1 repression of the *Pax6* gene [290].

### 3.17. Vsx1

The *Vsx1* gene belongs to the PRD class of homeobox genes. This gene family includes two genes in human and mouse genomes (*VSX1/Vsx1*, *VSX2/Vsx2*). *VSX1* gene contains five exons and encodes a protein with paired-type homeodomain. In addition to the paired homeodomain, VSX1 contains an additional 54 amino acid conserved regions, termed the CVC domain, located immediately adjacent to the C-terminus of the homeodomain [291,292,293]. Vsx1 can function as a transcriptional repressor [294].

During postnatal development, *Vsx1* is expressed in the presumptive INL, ganglion cells, and differentiating lens fibers, as well as in a few cells of the presumptive ONL (differentiating photoreceptor or horizontal cells). In the adult retina, *Vsx1* is most likely expressed in cone bipolar cells [295]. *Vsx1* is expressed in type 7 ON bipolar cells. Vsx1 is expressed only weakly in undifferentiated, presumptive neural retina of goldfish embryos and is then upregulated selectively in presumptive bipolar cells at early stages of differentiation before decreasing to an intermediate level, which is maintained in the differentiated adult retina. After completion of retinal lamination, *Vsx1* expression is restricted to cells occupying the INL and to postmitotic, differentiating progenitor cells in the growth zone at the peripheral retina, where neurogenesis continues throughout life [291,296,297]. Vsx1 is known to mark the largest subsets of cone bipolar cells [298,299,300]. In Xenopus, *Vsx1/Xvsx1* expression is maintained in retinal progenitors and in a peripheral region of the ciliary marginal zone, while in the central retina, it becomes restricted to differentiated bipolar cells [301]. It was found that *Xvsx1* is initially transcribed but not translated in early retinal progenitors. Its translation requires cell cycle progression and is sequentially activated in bipolar cells [302].

Bipolar cell fate was not altered in the absence of *Vsx1* function and expressed the pan-bipolar marker Chx10. The specification, number, and gross morphology of the subset of on-center and off-center cone bipolar cells were also normal in mutant mice. However, the terminal differentiation of OFF-CB cells was incomplete [300]. *Vsx1* mutant retinal cells form but do not differentiate a mature cone bipolar cell phenotype. Electrophysiological studies demonstrated that mutant mice had defects in their cone visual pathway, whereas the rod visual pathway was unaffected [298]. Thus, Vsx1 is necessary for bipolar cell terminal differentiation and is required for the activation and repression of gene expression in OFF and ON bipolar cells, respectively.

Mutations in the *VSX1* gene for distinct inherited corneal dystrophies, posterior polymorphous dystrophy and keratoconus, have been identified. These patients were found to have abnormal function of the inner retina, detected with electroretinography [303]. The mutations in the homeodomain and critical CVC domain of the *VSX1* gene result in abnormal craniofacial features, absence of the roof of the sella turcica, and anomalous development of the corneal endothelium. This mutation also impacts the maintenance of cone bipolar cells of the visual system [304,305]. Nonetheless, the *VSX1* gene has not been definitively demonstrated to play a causal role in keratoconus [306,307]. *VSX1* mutations in humans have also been associated with visual signaling defects, and individuals with *VSX1* mutations have exhibited abnormal ERG b-waves consistent with a defect in retinal bipolar interneuron function. H244R mutation of *VSX1* was reported to be associated with selective cone ON bipolar cell dysfunction and macular degeneration in a family with posterior polymorphous corneal dystrophy [308].

Mice lacking *Vsx1* exhibit ERG b-wave defects, decreased OFF visual signaling, and a perturbation of directional selectivity, all of which are thought to arise from dysfunctional cone bipolar cell signaling [298,300]. A mouse model for *Vsx1* p.P247R did not have any corneal defects, but did exhibit an abnormal electroretinogram response [305].

### 3.18. Vsx2

The *Vsx2* gene belongs to the PRD class of homeobox genes. This gene family includes two genes in human and mouse genomes (*VSX1/Vsx1*, *VSX2/Vsx2*). Like *Vsx1*, the *VSX2/CHX10* gene contains five exons and encodes a protein with an octapeptide, homeodomain, CVC domain and OAR domain and functions as transcriptional repressor [309]. The CVC domain has been shown to be involved in ubiquitin-mediated control of VSX2 protein stability [310]. Additionally, the CVC domain is important for the strength of homeodomain-dependent DNA binding [311].

*Vsx2* is first expressed in the part of the optic vesicle that gives rise to the neuroblasts of the optic cup [312]. *Vsx2* is expressed in RPC during much of development and is important for the ability of RPCs to proliferate [313]. *Vsx2* is not found in the ciliary marginal zone of Xenopus tadpoles, whereas it is expressed in RPC in warm-blooded vertebrates [314]. *Vsx2* is required for progenitor cell proliferation and formation of bipolar cells. In mice and chicks, Vsx2/*Chx10* is absent from all postmitotic cells except bipolar interneurons and a subpopulation of Muller glia [315,316]. *Chx10* has also been implicated in G1-phase cell cycle regulation, and *Chx10* mutations may cause cells to lengthen their cell cycle time [317]. Mitf (a transcriptional factor forced by an RPE cell identity) is expressed ectopically in the *Chx10^or^*^-J/or-J^ neuroretina, demonstrating that *Chx10* normally represses neuroretinal expression of Mitf. Ectopic expression of Mitf in the *Chx10^or^*^-J/or-J^ retina diverges it to an RPE-like structure [318].

In humans, a single base substitution in the DNA-recognition helix of the *VSX2* homeobox, a deletion of the homeobox, and a mutation of the CVC domain cause microphthalmia. A number of mutations of the *VSX2* gene have been reported in patients from West and South Asia to be associated with autosomal-recessive anophthalmia/microphthalmia, with or without iris coloboma and other ocular anomalies. All affected individuals with homozygous mutations had defects of ocular tissues ranging from an abnormally small eye size to complete bilateral anophthalmia or severe microphthalmia [56,319,320,321].

Ocular retardation (Or) mice with a null mutation of *Vsx2/Chx10* have microphthalmia, thin retinas, and optic nerve aplasia. The loss of *Vsx2* leads to both reduced proliferation of RPC and the absence of differentiated bipolar cells [322]. Mutations in *Chx10* cause reduced RPC proliferation and an absence of bipolar cells. Vsx2 directly represses *Vsx1* transcription in the mouse retina, and *Vsx1* mRNA is upregulated in the RPCs of *Vsx2*-deficient mice and zebrafish embryos injected with *Vsx2* morpholino [323]. Overexpression of *Vsx2/Chx10* after transfection with plasmid expressing this gene promoted differentiation of non-photoreceptor neurons while inhibiting differentiation of photoreceptor cells in dissociated retina of 5-day-old chick embryos [324]. In *Chx10*-null ocular retardation mice (*Chx10*^or-J/or-J^), delay of the normal temporal expression of genes essential for photoreceptor disc morphogenesis led to failure of correct rod and cone outer segment formation in the *Chx10*^or-J/or-J^ mutant retina. In addition, the absence of *Chx10* appears to affect the development of late-born cells more than that of early-born cells, in that a low number of rods develop, whereas formation of ganglion, amacrine, and cone cells is relatively unaffected [325].

## 4. Innovative Approaches of Modern Genomics and Cell Technology for IRDs Diagnostics

Current areas of study include the genetic basis of the etiopathogenesis of retinal diseases and how to improve the methods of genetic diagnostics that underlie technologies for directed editing of the differentiated cell genome and reprogramming [10]. Knowledge about the molecular aspects of inherited eye diseases, and particularly IRDs has increased tremendously in recent years. However, the main problem in their diagnostics is a very high degree of clinical and genetic heterogeneity. There are examples of significant differences between phenotypes (e.g., age at onset) in IRDs that carry the same mutations, both within and between families. The development of new technologies is aimed at finding the causative genetic variants, with special attention paid to next-generation sequencing and genome-wide SNP homozygosity mapping, which can combine molecular diagnostics and retinal disease gene identification [320,326]. Important aspects of disease diagnostics are the functional assessment of rare variants with RNA and protein effects that can only be predicted in silico and the assessment of putative splice defects. Very useful references on several key databases that provide extensive information about gene variants, phenotypic information that accompanies variants, and information about genotype–phenotype correlations were summarized [8]. Most databases make use of Human Genome Variation Society (HGVS) nomenclature to describe variants in a consistent manner (https://varnomen.hgvs.org). To study the mechanism of variable expression, there are high hopes for genome-wide analysis techniques such as whole exome sequencing (WES) and whole genome sequencing (WGS) [327,328].

The development of modern technologies of DNA sequencing of single cells and genome sequencing has allowed experiments to be carried out at the level of individual differentiating RPE and retinal cells from human fetuses [329,330]. Spatiotemporal characteristics and dynamics of the appearance and onset of functioning of one or another cell type are now obtained using single-cell RNA sequencing (RNA-seq), which makes it possible to obtain specific cellular “transcriptome landscapes” of retinas taken at weeks 5–24 of human fetal development [331]. This method, along with bioinformatics analysis and immunochemistry, make it possible to isolate the transcriptomes of individual retinal cells sequentially differentiating into the main cellular types—photoreceptors, interneurons, ganglion cells, and Muller glia cells—along with the specialization of RPE cells, which occurs in accordance with human neural retina maturation. Data obtained were clustered into classes corresponding to certain cell types, which, in turn, were identified on the basis of markers that were previously determined and turned out to be close to the cellular types of retinas of mice and humans [331].

Bioinformatics analysis of the main distinguished transcription factors shows that they have the highest activity in regulatory networks providing differentiation of RPE and neural retina separately. It was found that the RPE and retina of the human eye have differences in the networks of genetic expression quite early [332]. A set of expressed genes specific to prospective neural retina was highly characteristic of the development of the nervous system, while RPE cells expressed genes associated with retinol metabolism [333]. In addition to substantial detailing and refinement of existing information, this made it possible to come closer to understanding the functional role of the expression of certain genes in the development of these two major eye tissues. Until now, this has been a bottleneck against a backdrop of the rapid accumulation of data on the work of certain genes in the retinas of developing vertebrates, and particularly humans.

The genomics approaches clarified the genetic signature and time of appearance of certain cell types on the basis of sequencing of isolated cells [334]. It turned out that the owners of a particular phenotype in the prospective retina for a specific set of transcription factors may be present much earlier than the dates previously known for them, albeit in a small number, and only much later to reach peak expression. For example, individual ganglion cells, according to the profile of specific expression, are detected already at 5 weeks, but their maximum population is formed only at 8 weeks. It was also found that early RPC express genes specific for mature retinal rods, but not cones. A greater similarity in the set of transcription factors between cells of the late fetus and adult compared with the early stages of gestation is expected to be revealed. A comparison was made with the known data for the retina of macaque and mouse. It is interesting that a greater similarity in the transcriptome of fetal cell types of the retina of humans and mice than humans and monkeys revealed [332]. Analysis of the transcriptome of RPE cells in their development in humans revealed a steady decline in *PAX6* expression and an increase in the expression of visual cycle genes such as *RPE65* and *LRAT*. Moreover, proliferating RPE cells were detected at 5–6 weeks, and rhodopsin expression at 24 weeks, although other genes associated with the visual cycle had expression at 13 weeks of development. Thus, determining the cells of the developing human retina based on their individual transcriptome provided information on both transcription factors expression and structural proteins. These data allowed us to come closer to understanding the functional role of molecular participants in the regulation of spatiotemporal dynamics of gene expression [331]. Information has been made available on the development of a detailed gene map of the human retina as part of the Human Cell Atlas Project, a global project to create reference maps of all human cells [335]. The genetic sequences of more than 20,000 individual cells were examined in order to develop a gene profile of all major retinal cell types to support the function and stability of the retina. Constantly enriching the data on the human retina genetic map will help us to understand the molecular profile of individual retinal cell types that enable cells to keep functioning and contribute to healthy vision, and will help us to study how those genes impact different kinds of cells. From the other side, a molecular atlas of healthy cells with cells from retinal diseases and across different stages of human development will help us to understand the factors and signals that cause cells to stop functioning and lead to vision loss and blindness [335]. Variations in gene expression coupled with the gene’s biological sources provide insight into the complex pattern of retinal gene expression that underlies specific phenotypic tissue differences, and physiologic conditions within a tissue and between normal and disease states [55]. Large-scale genetic screening of IRDs patients using whole exome/genome sequencing will continue to improve the efficiency of diagnosis through identifying novel disease-associated genes and their isoforms associated with inherited retinal degeneration [327,328].

## 5. Gene-Based and Cellular Technologies in the Treatment of Inherited Retinal/Eye Diseases

Experimental approaches of modern genomics for the treatment of retinal pathologies associated with impaired functions of functionally significant genes are developing rapidly. Today, this work is based on the achievements of modern cellular technologies and closely related genetic approaches that are developed primarily on animal models in vivo and in vitro or cell cultures of human eye tissues [11,336,337,338,339].

In this review, we consider a few examples of experimental and clinical work that is being done to make up for significantly reduced or lost vision in humans. Genetically determined diseases of the retina associated with mutations in genes that regulate eye development are an incentive to improve and apply technologies for targeted genome editing. The use of directional genome editing using clustered regularly interspaced short palindromic repeats (CRISPR-Cas9) systems is a promising area in biomedicine and ophthalmology for the treatment of inherited eye diseases This approach is being worked out to eliminate genetic defects leading to retinal development pathologies and loss of vision, such as RP, in experimental animals [7,340] Earlier, there was evidence of successful gene therapy for LCA in mice, using an adenovirus to deliver the gene construct as a carrier [341]. Currently, work is actively ongoing in this area, using gene constructs to eliminate defects in the neural tissue of the eye. It is known that one of the reasons for the development of IRDs is dysfunction of RPE cells. The development of degenerative diseases of the retina—RP, and LCA—is associated with mutations in the retinal pigment epithelium 65 (*RPE65*) gene, which is specifically expressed in RPE. Clinical studies have begun on the treatment of these diseases by introducing the functional *RPE65* gene [336].

For the first time, an attempt was made to treat RP in a mouse model using CRISPR-Cas9 technology. RP is a congenital disease, genetically variable, caused by disorders in more than 60 genes with 3000 mutations, which is reason to act with the help of gene therapy. It is known that RP is caused by the gradual death of rods in the retina (which not only are responsible for night vision, but also structurally and trophically support cones) in the beginning of disease, and then cones. Using a mouse eye model, CRISPR sgRNA, which targets the retinal transcription factor neural retina leucine zipper (Nrl), responsible for the maturation of rods in the developing retina and their maintenance in the adult retina, was developed [9]. *Nrl-*targeted sgRNA and Cas9 endonuclease were delivered directly to the retina of mice using two adeno-associated viral (AAV) vectors. The role of Nrl is to maintain the functioning of rods, and its elimination leads to a partial conversion of rods into cones and, much more significantly, a condition supporting the viability of cones. The proposed technology of direct delivery of the AAV design to mammalian and human retinas, mediated by CRISPR-Cas9, as shown by experiments on mice, works with different forms of retinal degeneration caused by PR. Injections were given to 2-week-old mice, and observations were made 3–4 and 6 months later. As a result, at first there was a significant decrease in *Nrl* expression, and then downregulation of rod genes and upregulation of cone cells, as well as inhibition of the death of cone cells [9]. However, in this case, which was an absolute success, where the experiments were carried out in a mouse model, with not one but 67 genes involved in the development of pathology, important questions remain. Answers to them must be obtained before the method is used in practice. In particular, it is still unknown to what extent the data can be extrapolated to the human retina, and in particular, if long-term expression of Cas9 is harmful to mammalian and human retinal cells.

Despite the advances in CRISPR-Cas9 technology, there are some limitations that should be taken into consideration and overcome for successful therapy. It is known that Cas9 induces double-strand breaks on sequences that differ by one or more base pairs from the gRNA sequence, which is the reason for the off-target therapeutic effects [342]. Strategies to overcome this problem include improving the specificity of the Cas9 molecules, that guide RNA design and the improving the delivery methods, because most *Cas9* genes are too big to be delivered by a single AAV vector. Attempts are being made to solve this problem, or at least find approaches to solving it [343,344]. Smaller Cas9 molecules have been discovered that are small enough to be packaged within an AAV [345]. It opens up the prospect of directly administering the *Cas9* gene into the retina for genome editing in vivo.

With the existing shortcomings, gene therapy holds great promise for developing a cure. The first gene therapy drug targeting IRDs caused by mutations in a single gene, Luxturna™, an AAV2/2 vector carrying the *RPE65* gene for the treatment of *RPE65*-associated RP or LCA type 2 (voretigene neparvovec-rzyl; Spark Therapeutics, Inc., Philadelphia, PA, USA), has recently been approved. This product delivers a normal copy of the *RPE65* gene to retinal cells for the treatment of biallelic *RPE65* mutation–associated retinal dystrophy, a blinding disease [11].

Thanks to the successes of molecular genetics and experimental embryology, a new field of medicine has emerged, regenerative medicine, which combines molecular biological, pharmaceutical, cellular and tissue technologies to stimulate reparative processes in pathologies and retinal injuries [346]. The use of cellular technologies consists of attempts to experimentally reprogram source cells (embryonic stem cells, embryonic PRC, induced pluripotent cells (iPSC), poorly differentiated cells), obtain organoids based on them, and develop methods of transplantation, in an attempt to replace them with the help of pathology or damaged tissue of the eye (retina, RPE) [339,347,348,349,350].

Some of the developed approaches are aimed at creating specific models of autologous cells and tissues with corrected genotypes for use in cell replacement therapy and tissue engineering [351]. The most important achievement in technologies for producing pluripotent cells from differentiated cells of an adult organism was made during the last decade. The iPSCs were obtained from fibroblasts of human and mouse skin using transduction of four transcription factors: Oct3/4, Sox2, Klf4, and c-Myc [352]. In the field of ophthalmology, iPSC technology is used in the development of methods for treating retinal dystrophy. So, there were reports of decreases in the degenerative processes of the RPE of the monkey following the placement of pigmented epithelial cells obtained using iPSC technology from skin fibroblasts on the retina [353]. In 2013, Japan began clinical trials using iPSC-derived RPE cells to treat AMD, in which the destruction of central retinal cells occurs [354]. The application of stem cell technology to generate photoreceptor precursor cells from patient somatic cells, which can subsequently be used for RNA and protein studies, has been described [8,355]

Approaches to creating a functioning human retina with restoration of three-dimensional structure in vitro are being developed [355,356]. One study reported on the possibility of obtaining a fragment of the neural tissue of a human retina from human iPSCs in 3D in vitro (Johns Hopkins University School of Medicine, Baltimore, MD, USA) [356]. The emerging tissue, planted in cups on a special matrigel and in a medium providing neural differentiation, had certain developmental dynamics similar to ontogeny in humans and was maintained in vitro for 200 days. In this retina, all seven types of retinal cells expressing specific molecular markers were identified morphologically. The first steps were to obtain morphologically mature photoreceptors from human iPSCs that had all the attributes of functional specialization: the inner and outer segments of the photoreceptor cell. These cells contained optic opsin pigments, transmembrane proteins responsible for transmitting a signal from captured light photons to a physiological response, which was recorded by electrochemical assay [356]. This work demonstrated the possibility of further growth and differentiation of photoreceptors obtained from human iPSCs, followed by the formation of tissue containing cells not only in the desired ratio and orientation, but also with functional activity. In that study, the autonomous mechanism of the output of artificially grown retinal cells into differentiation was a surprise. It was probably due to the presence of as yet unknown endogenic instructions accompanying the development of the retina in the absence of other eye tissues. The results of this study are a definite achievement, since so far it has not been possible to bring the resulting cells into deep differentiation physiologically close to a complete retina. The authors of that study warned that a significant amount of time is still needed until such developed cellular products can be used in clinical practice.

The other study describes the failure of stem cell transplant attempts in three patients, older women with AMD. Adipose cells taken from patients were used to obtain stem cells, but they were not fully tested for their properties and possible pathogenicity. Transplantation of these cells to the AMD patients led to serious complications and loss of vision in all three cases. The article also notes that this case is only the tip of the iceberg, since it is known that transplants of unverified cellular material are carried out in 600 US clinics in the same way. The result of AMD treatment is still small, the absence of further visual impairment for a short period of time, about 1 year [351].

In all attempts at transplantation with iPSCs or other sources of cells, cell aggregates, or fragments of retinal tissue, the most difficult and most important aspects (since we are talking about sensory tissue) are successfully integrating the transplants into the tissues of recipients and establishing the correct contact both with other cells and with the visual analyzer [346]. The results of attempts related to cell transplantation are very modest due to the complexity of the structure of the retina, the occurrence of gliosis, and the formation of a scar (in the case of proliferative or diabetic retinopathy) [357].

## 6. Pharmacologic Neuroprotection and Activation of Endogenous Cell Potential as an Alternative to Genetic Engineering Methods

Significant progress in the field of pharmacologic neuroprotection for mammals and humans is driven in large part by advances in approaches aimed at suppressing gliosis and fibrosis and activating internal regenerative reserves, which remains an urgent problem [358]. The main directions in studies of the regenerative potential of the retina are the search for exogenous and endogenous cellular sources to replenish the pool of cells and the development of pharmacologic drugs for the treatment/restoration of the retina, in situations where this may be feasible [359].

Despite the high incidence of retinal diseases and the complexity of mechanisms involved, several promising neuroprotective treatments provide hope to prevent blindness [360]. Since transplantation of exogenous cellular sources for the retina is not always successful, endogenous stimulation has many advantages. In this regard, the search continues for opportunities to stimulate potential endogenous cellular sources, directing control of the differentiation of cells to restore tissue and the normal functioning of the organ of vision [360,361,362]. Studies show that in damaged neurons of the retina, activation of the transcription program is observed, which promotes initial recovery and partial restoration of function. Existing strategies for the use of pharmacologic antioxidants for the pathologies of neural tissues are directed to allow them to activate their own regenerative potential and antioxidant cell systems and maintain neurogenesis. The *ADNP* gene is one example of homeobox genes used for neuroprotection. This approach uses stimulation of secretion of the ADNP-derived octapeptide NAP, which enhances autophagy and protects ganglion neurons in conditions of retinal ischemia and optic nerve crush [65,71].

In mammalian and human retinas, cells with potential properties of stem cells are Muller glia and RPE, which, with damage or pathology, can dedifferentiate to form a population of progenitor cells developing in the neuronal direction [363]. The data on the hypomethylated status of DNA in fish Muller glia indicate the possibility that these cells can undergo phenotype change under the influence of internal and external factors, and can take on roles in epigenetic reprogramming mechanisms and regenerative responses [364]. In the retinas of mice damaged as a result of the action of neurotoxin, it is possible to stimulate cell division of Muller glia and neurogenesis as a result of overexpression of the *Ascl1* gene [365]. Muller glia cells of the adult retina, as well as the RPE, in the in vitro system are characterized by the activity of transcription factors characteristic of RPC PAX6, SOX2, and CHX10 [366]. Various experimental approaches to the regulation of proliferation and activation of the neurogenesis program in Muller glia cells are based on stimulating proliferation with exogenous signaling proteins Wnt, Notch, and Hedgehog [367] or activation of endogenous transcription factors [368,369]. Exogenous neurotrophic factors can be delivered to neuronal tissue to promote neuronal survival and regeneration [370,371]. In a number of studies, it was shown that ciliary neurotrophic factor (CNTF) [372,373], pigment epithelium derived factor (PEDF) [374], and hepatocyte growth factor (HGF) can act as stimulants of reparative processes [375].

Growth factors can be secreted by glial cells, including microglia and macroglia, like the retinal astrocytes and Muller glia. Muller glia release factors such as PEDF, transforming growth factor β (TGFβ), vascular endothelial growth factor (VEGF), BDNF, and NGF [373]. In retinal pathologies, the release of these growth factors declines, confirming the hypothesis that supporting these factors to normal levels enhances neuronal survival. Some factors, such as BDNF, are critical for synaptic plasticity, i.e., the strengthening of synaptic transmission after the activation of neuronal firing, in addition to promoting neuron survival [376,377]. Muller glia cells can be induced to de-differentiate and be brought into a pluripotent state in vitro when incubated with extracellular vesicles (exosomes) obtained from ESCs, which may indicate the possibility of their activation to restore the retina in vivo [378].

It is known that application of neurotrophic factors for neuroprotection has significant limitations due to the fact that most these factors are hydrophilic and monomeric or dimeric proteins. This can be explained by their inability to easily pass through the blood–brain, blood–retinal, or blood–cerebrospinal barrier. In addition, it was noted that neurotrophic factors are characterized by short in vivo half-lives and poor pharmacokinetics [379]. The local delivery of neuroprotective factors to the eye avoids some of these limitations.

Another approach that is still being worked out on experimental animal models involves the use of synthetic analogues of angiogenesis stimulants such as epoxyeicosotrienic acids (EETs) in the retina. It is known that vascular endothelial cells when damaged produce EETs that stimulate angiogenesis by activating the synthesis of VEGF [380].

An alternative neuroprotective strategy is based on the activation of potential endogenous mechanisms through the use of neurotransmitters and their antagonists, and targeting of the corresponding receptors [381,382]. For example, this strategy involves the use of purinergic P2X receptor antagonists and A3R adenosine receptor agonists as neuroprotectors and regulators of retinal repair [383]. Stimulation of RPE cells by a nicotinic acetylcholine receptor agonist also contributes to the activation of Muller glia proliferation and neural differentiation in adult mice [384].

Among the many neuroprotective drugs in AMD and glaucoma are targets of survival pathways, including bile acids such as ursodeoxycholic acid (UDCA) and its taurine-conjugated derivative tauroursodeoxycholic acid (TUDCA), neurotrophic factors, and therapies that target retinal dopamine [359]. The success of these approaches depends largely on the delivery of dopamine-targeted treatments to the retina, focusing on sustained release, microneedles, and molecules that are specifically tagged for retinal delivery.

## 7. Conclusions/Insights

We have shown that most of the key homeobox genes are involved in both retinal development and IRDs. Homeobox genes perform many functions in the retina, so their mutations are characterized by diversity among different diseases. The range of phenotypes associated with variants in regulatory genes cause phenotypic heterogeneity [385]. IRDs have in some cases multigenic etiologies, which make them difficult to treat [7]. Besides, normal eye tissue interactions and metabolic disorders, and vascular changes might also be primarily causative of pathological degenerative changes in the retina appearing as secondary effects. All of these multiple factors should not be overlooked, particularly if they become altered under the influence of changed pathological conditions. Comprehensive characterization of the pleiotropic genetic effects could improve our understanding of the etiologies of eye diseases and their mechanisms [386], as the choice of strategies requires considering the systemic nature of these retinal pathologies [387].

Advanced sequencing methods and the use of cell type-specific transcriptome and proteome and high-throughput screening systems of phenotypic and pleiotropic genetic effects analysis are aimed at identifying target genes associated with inherited diseases and provide prerequisites for better diagnostics, appropriate therapy technologies, and pharmacologic retinal protection [331,335,388,389].

Much of the success in using iPS technologies and gene modifications, as we see, has been with research conducted on laboratory animals, but translating the results into medical practice is more complicated. The iPCS-based models provide many benefits for modeling retinal pathologies, but certainly have some limitations, since it is not always possible to draw a direct connection from model animals to human IRDs [390,391,392]. In attempts to transmit data from experimental animal models to the human eye, it is important to take into account species barriers and tissue-specific features of regulating the expression of regulatory transcription factors, among which are the homeobox genes [391]. The patient-specific iPSC for deriving retinal lineages are a powerful alternative tool for discovering new disease-causing mutations and developing personalized cell therapy [393].

Cellular and tissue-like 3-D-models of inherited eye diseases in general, and IRDs in particular, in combination with genome-editing technologies such as CRISPR-Cas9, will further advance our understanding of the function of homeobox genes, their place in the regulatory hierarchy of underlying retinal pathogenesis, and how their pathogenic alleles result in retinal disorders in order to create therapies for IRDs. The use of genome-editing technology can lead to major breakthroughs in therapy, but using it safely is impossible without knowledge of the genetic nature of the cell, the characteristics of tissues, cellular responses, and compliance with ethical and social factors. The attitude of the International Society for Stem Cell Research (ISSCR) toward research based on the use of the listed gene technologies is ambiguous [8,351,355,356]. In parallel with the study of functions and their place in the genetic regulation program, a large-scale search for molecular targets is being carried out [327,328,394].

Optimization of viral delivery methods is aimed at revealing new vectors with improved characteristics and using less invasive routes of administration (e.g., intravitreal or suprachoroidal injection) for more efficient transduction of target tissues [345,346]. Efforts in the area of nonviral delivery also continue with the EyeCET electroporation technology currently being evaluated [11]. One way to maintain retinal function may be to attempt to compensate gene function by activating synergistic genes.

Neuroprotective therapeutic approaches have the potential to prevent vision loss effects with eye diseases, but despite encouraging preclinical data, the translation of neuroprotective approaches into the clinic has been fraught with failure [360]. The problems of using latent exogenous cells with stem properties for neuroprotection and restoration of the retina in humans are still far from being resolved, which is due to both the complexity of the highly organized structure of the retina and its persistent tendency to form scar tissue [346]. Currently, cell transplantation and pharmacologic antioxidant drugs still do not allow the structure of the retina in adult mammals and humans to be fully restored in vivo. These approaches make it possible to delay the death of the retinal cells and maintain cell viability for some time [347,349,395].

Activation of endogenous retinal regenerative potential seems to be the most promising approach for regenerative medicine and ophthalmology [396,397,398]. It is obvious that successes in therapy for human degenerative retinal diseases are in line with the efficiency and safety of the biomedical approaches, and thorough preclinical studies are required before treatments to avoid potential failure.

## Figures and Tables

**Figure 1 ijms-21-01602-f001:**
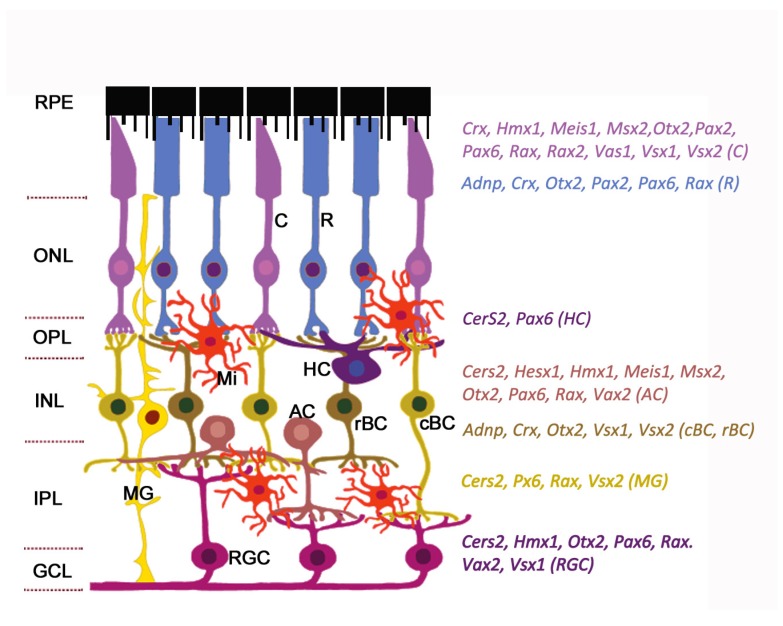
Expression of homeobox genes in the adult mouse retina. The retinal architecture is detailed in the review. The cell-specific expression of genes is detected by single-cell RNA sequencing (data from https://eyeintegration.nei.nih.gov). The only genes which are known to associate with eye/retinal malformations in humans are shown. On the left: Retinal layers; on the right: homeobox genes indicated in the same color as the cell types expressed them. Abbreviations in parentheses show corresponding cell types. RPE, retinal pigment epithelium; ONL, outer nuclear layer; OPL, outer plexiform layer; INL, inner nuclear layer; IPL, inner plexiform layer; GCL, ganglion cell layer; C, cone; R, rod; HC, horizontal cell; AC, amacrine cell; cBC, cone bipolar cell; rBC, rod bipolar cell; MG, Muller glia; RGC, retinal ganglion cell; Mi, microglial cell. Modified from [13]. License to reproduce: https://creativecommons.org/licenses/by/4.0/.

**Table 1 ijms-21-01602-t001:** Properties of human homeobox genes associated with inherited retinal diseases (IRDs) and secondary retinal malformations.

Gene	Full Name	Synonyms	Ensembl ID	Location	Transcripts (Total/Protein-Coding) (Ensembl)	Homeobox Class	Expression in the Retina	
							RNA (Microarray) (BioGPS)	Protein (ProteomicsDB)
*ADNP*	Activity-dependent neuroprotector homeobox	*ADNP1*, *KIAA0784*	ENSG00000101126	20q13.13	9/8	ZF	20	4
*ALX1*	Aristaless-like homeobox 1	*CART1*, *HEL23*	ENSG00000180318	12q21.31	1/1	PRD	3	0
*CERS2*	Ceramide synthase 2	*LASS2*, *TRH3*, *TMSG1*	ENSG00000143418	1q21.3	14/9	CERS	40	3
*CRX*	Cone-rod homeobox	*CORD2*, *OTX3*, *LCA7*	ENSG00000105392	19q13.33	7/4	PRD	43	18
*HESX1*	Homeobox gene expressed in embryonic stem cells 1	*ANF*, *CPHD5*, *RPX*	ENSG00000163666	3p14.3	4/4	PRD	4	–
*HMX1*	H6 family homeobox 1	*Nkx5-3, H6*	ENSG00000215612	4p16.1	2/2	ANTP	13	0
*LMX1B*	LIM homeobox transcription factor 1 beta	*NPS1*, *LMX-1.2*	ENSG00000136944	9q33.3	4/4	LIM	3	0
*MEIS1*	Myeloid ecotropic insertion site homeobox 1	–	ENSG00000143995	2p14	17/6	TALE	6	2
*MSX2*	Muscle segment homeobox 2	*MSH*, *HOX8*	ENSG00000120149	5q35.2	2/2	ANTP	4	0
*OTX2*	Orthodenticle homeobox	–	ENSG00000165588	14q22.3	11/11	PRD	4	12
*PAX2*	Paired box 2	–	ENSG00000075891	10q24.31	9/6	PRD	3	0
*PAX6*	Paired box 6	–	ENSG00000007372	11p13	82/57	PRD	75	5
*RAX*	Retina and anterior neural fold homeobox	*RX*	ENSG00000134438	18q21.32	4/3	PRD	4	0.52
*RAX2*	Retina and anterior neural fold homeobox 2	*RAXL1, QRX*	ENSG00000173976	19p13.3	2/2	PRD	13	5
*VAX1*	Ventral anterior homeobox 1	–	ENSG00000148704	10q25.3	2/2	ANTP	–	–
*VAX2*	Ventral anterior homeobox 2	–	ENSG00000116035	2p13.3	3/3	ANTP	3	–
*VSX1*	Visual system homeobox 1	*RINX*, *KTCN*	ENSG00000100987	20p11.21	7/6	PRD	3	–
*VSX2*	Visual system homeobox 2	*CHX10*, *HOX10*, *RET1*	ENSG00000119614	14q24.3	1/1	PRD	4	4

Ensembl ID: data from Ensembl, version 98, September 2019, for human genes; RNA expression: microarray data from BioGPS; Protein expression: estimated protein expression log10 (ppm) (according to ProteomicsDB). “–“ synonyms are unknown. Homeobox classes: ANTP, named for the *Antennapedia* (*Antp*) gene of Drosophila; CERS, CERamide Synthase; LIM, named by the initials of the three homeodomain proteins Lin11, Isl-1 and Mec-3; PRD, PaiReD; TALE, Three Amino acid Loop Extension; ZF, Zinc Finger.

**Table 2 ijms-21-01602-t002:** Cell-type specificity and function of homeobox genes expressed in neural retina and associated with inherited eye diseases in humans.

Gene	Full Name	Expression in the Retina/RPE		Cell Functions	Disease or Syndrome	
		Single-Cell RNAseq (mouse)	RT-PCR, IHC		Disease/Syndrome (OMIM#)	Ocular Manifestations in Humans
*ADNP* (611386)	Activity-dependent neuroprotector homeobox	BC, rods	INP, OPL	Neuroprotection, promotion of neuronal growth and differentiation, autophagy	Helsmoortel–van der Aa syndrome (615873)	Macular laminations, foveal hypoplasia, cone degeneration
*ALX1* (601527)	Aristaless-like homeobox 1	Early RPC (E14)	Retinal margin	Cranial neural crest migration and differentiation, retinogenesis	Frontonasal dysplasia, type 3 (613456)	Anophthalmia, microphthalmia, optic nerve hypoplasia, coloboma
*CERS2* (606920)	Ceramide synthase 2	AC	All neurons, MG	Apoptosis, inflammation, signal transduction, lipid metabolism	Rhegmatogenous retinal detachment (Stickler syndrome) (PS108300) AMD	Retinal detachment, photoreceptor degradation, lipofuscin granules, RPE atrophy
*CRX* (602225)	Cone-rod homeobox	BC, cones, rods	RPC	Development and maintenance of Phr, renewal of Phr disks	Cone-rod retinal dystrophy type 2 (120970) Leber congenital amaurosis type 7 (613829) Dominant retinitis pigmentosa (268000)	Cone-rod retinal dystrophy, widespread retinal pigmentation, chorioretinal atrophy, attenuated retinal vessels, cystoid macular edema
*HESX1* (601802)	Homeobox gene expressed in embryonic stem cells 1	–	Early RPC, AC	Maintenance of stemness, neural cell determination	Septo-optic dysplasia (182230) Ocular coloboma (120200)	Optic nerve hypoplasia, hypoplastic optic discs, microphthalmia
*HMX1* (142992)	H6 family homeobox 1	AC, cones, RGC	Optic vesicle, early RPC, nasal RGC	Neurogenesis, nasotemporal patterning of retina	Oculoauricular syndrome (612109)	Microphthalmia, ocular coloboma, RPE anomalies, rod-cone dystrophy, macular hypoplasia
*LMX1B* (602575)	LIM homeobox transcription factor 1 beta	–	Periocular mesenchyme	Optic cup morphogenesis, nasotemporal patterning of retina	Nail-patella syndrome (161200) Primary open-angle glaucoma (137760)	Isolated optic disk excavation, ocular hypertension
*MEIS1* (601739)	Myeloid ecotropic insertion site homeobox 1	RPC, AC, cones, PhR precursors	–	Control of cell cycle in RPE, dorsoventral and nasotemporal patterning of retina	Microphthalmia (?)	Microphthalmia
*MSX2* (123101)	Muscle segment homeobox 2	AC, cones	–	EMT, suppression of transcription, apoptosis, RGC commitment and differentiation	Craniosynostosis type 2 (Boston-type) (604757)	Chorioretinal coloboma
*OTX2* (600037)	Orthodenticle homeobox 2	Late RPC, AC, rods, cones, BC, RGC (E14-P0)	RPC, RPE, RGC, PhR, BC	RPE specification, differentiation of photoreceptors and bipolar cells	Microphthalmia, syndromic type 5 (610125) Early-onset retinal dystrophy (610125)	Microphthalmia or anophthalmia, ocular coloboma, retinal dystrophy, optic nerve dysplasia
*PAX2* (167409)	Paired box 2	Rods, cones (E18)	Ventral optic vesicle, optic fissure, optic stalk, astrocytes	Inhibition of neurogenesis, induction of gliogenesis, axon guidance	Papillorenal syndrome (120330)	Retinal and optic nerve colobomas, optic disc dysplasia or hyperplasia, microphthalmia, gliosis of optic nerve, abnormal retinal vessels, chorioretinal degeneration
*PAX6* (607108)	Paired box 6	Early and late RPC, AC, cones, rods, RGC, MG	RPC, AC, RGC, HC	Maintenance of RPC multipotency, proliferation and differentiation of RGC and RPE, differentiation of HC	Foveal hypoplasia 1 (136520) Optic nerve hypoplasia (165550) Coloboma of optic nerve (120430)	Optic nerve hypoplasia, coloboma of optic nerve, microphthalmia
*RAX* (601881)	Retina and anterior neural fold homeobox	Early and late RPC, AC, rods, cones, RGC	RPC, rods, cones, MG	Proliferation of RPC, differentiation of PhR	Microphthalmia, isolated, type 3 (611038)	Microphthalmia (retinal size reduction)
*RAX2*	Retina and anterior neural fold homeobox 2	–	RPC, cones	Differentiation of cones	Cone-rod dystrophy type 11 (610381); AMD type 6 (613757)	Progressive macular atrophy, pigment granularity of peripheral retina, mixed rod and cone dysfunction on electroretinography, atrophy of macular RPE, progressive attenuation of retinal vessels, macular degeneration
*VAX1* (604294)	Ventral anterior homeobox 1	Late RPC	Glia of the optic nerve, ventral optic stalk	Repression of retinogenesis, RGC axon growth and guidance, dorsoventral patterning of retina	Microphthalmia, syndromic type 11 (614402)	Bilateral severe microphthalmia, small optic nerve
*VAX2* (604295)	Ventral anterior homeobox 2	AC	Cones, RGC	Repression of retinogenesis, dorsoventral patterning of retina, differentiation of cones, organization of nerve fiber layer within retina	Cone dystrophy, coloboma	Macular and cone degeneration
*VSX1* (605020)	Visual system homeobox 1	BC, cones	Cone BC (cBC), RGC	cBC differentiation	Craniofacial anomalies and anterior segment dysgenesis syndrome (614195)	Macular degeneration, BC dysfunction
*VSX2* (142993)	Visual system homeobox 2	Early and late RPC, BC, cones	RPC, all BC, MG	RPC proliferation, neuroretina specification	Microphthalmia, isolated type 2 (610093) Microphthalmia with coloboma type 3 (610092)	Microphthalmia, coloboma

AC, amacrine cells; BC, bipolar cells; HC, horizontal cells; MG, Muller glia; PhR, photoreceptor; RGC, retinal ganglion cells; RPC, retinal progenitory cells; RPE, retinal pigment epithelium; OPL, outer plexiform layer; IPL, inner plexiform layer; IHC, immunohistochemistry. Single-cell RNAseq (mouse) – data from https://eyeintegration.nei.nih.gov [77] showing cells that expressed high levels of transcripts (decile of mean gene expression = 8–10) in prenatal and postnatal mouse retina. “–“ data are not found.

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
