# Peer review of "Inherited Eye Diseases with Retinal Manifestations through the Eyes of Homeobox Genes"

_ijms, 2020, doi:10.3390/ijms21051602_

Round 1

Reviewer 1 Report

The authors reviewed the recent literature on inherited retinal diseases, associated genes ad their role in retina development as well as current approaches and trends to treat inherited retinal diseases and injuries. This is a very broad review of the field. 

page 1, line 39. The authors wrote "loss-of-gain function". It should be probably "loss-of-function". The authors should provide an illustration of retina layers that support their description in section 2. retinal organization. Gene names in tables and in the entire manuscript should be written in italics. The article needs an extensive English language editing.

Author Response

Response to Reviewer 1 Comments

Point 1: page 1, line 39. The authors wrote "loss-of-gain function". It should be probably "loss-of-function".

Response 1: page 1, line 37. "loss-of-gain function" is change to “loss- or gain-of-function”

Point 2: The authors should provide an illustration of retina layers that support their description in section 2: retinal organization.

Response 2: The figure of the retina with homeobox genes expressed is provided (Figure 1, page 2).

Point 3: Gene names in tables and in the entire manuscript should be written in italics.

Response 3: We rewrote gene names in italics and capitalised human genes and proteins.

Point 4: The article needs an extensive English language editing.

Response 4: We used English editing service provided by MDPI to check English of our manuscript (english-edited-16444) and corrected the text according their recommendations (please see the attachment for English-Editing-Certificate).

Reviewer 2 Report

The authors have written a comprehensive review concerning the role of homeobox genes in retinal diseases. The work put into the review is extensive and it is highly referenced, with 404 citations. It is regrettable that there are no illustrations to help the reader (for example, graphic representations of the eye and where each gene has a role).

My main general comment, is that the authors seem to have confused inherited retinal diseases (IRDs) with pan-retinal or, in some cases, pan-ocular diseases. IRDs are a specific group of diseases that are characterized by loss of photoreceptors leading to, in most cases, progressive visual loss (although stationary forms do exist). However, in this review, the authors discuss genes that are not related to IRDs, such as for example, Lmx1B, which leads to anterior segment malformations or optic neuropathy. To centre the review on IRDs, would require the removal of a large amount of text, which seems a shame when we consider the work put in. Therefore, it would be likely more appropriate to alter the title and the text to indicate that not only IRDs are treated in the review. Maybe, for example, “Inherited ocular diseases through the eyes of homeobox genes”.

My specific comments are as follows:

  1. Overall, the review is comprehensible however there are grammatical English errors so I would recommend careful revision by a native English speaker. These were too numerous to cite but the authors could keep in mind that if a noun becomes an adjective it should no longer be in plural, for example “retinal cells” as opposed to “retinal cell types”.
  2. Line 39, Page 1, there seems to be a typographical error. What do the authors mean by “loss-of-gain function”? The term is usually “loss-of-function” or “gain-of-function”. Alternatively, perhaps they mean “loss- or gain-of-function”?
  3. Line 41, Page 1, or Line 14, Page 2, the authors use “ganglionic cells” or “ganglious cells”, respectively. Please homogenise to “ganglion cells” throughout the manuscript.
  4. Line 18, Page 2, “sensor neurons” should be “sensory neurons”.
  5. Lines 34 & 35, Page 2, please change to “between bipolar and ganglion neurons”.
  6. Line 46, Page 2, what do the authors mean by “microvessels of the wall cells”? Please correct.
  7. Line 1, Page 3, please change to “strict spatial and temporal regulation”.
  8. Line 24, Page 3, please change “as differentiation retina proceeds” to “as retinal differentiation proceeds”.
  9. Line 41, Page 3, please change “conservative” to “conserved”.
  10. Table 1, Please also define the homeobox classes in the foot notes of the table. These abbreviations are defined in the text but often after the table.
  11. Line 41, Page between tables 1 and 2, it is written that patients with ALX3 mutations are “without significant eye manifestations”, so it is not clear why this gene is listed in table 1. However, in table 2 it is listed that it is associated with micropthalmia. Please homogenise.
  12. Table 2, The title needs to be changed as IRDs are not associated in all cases. Furthermore, the title of the last column entitled “Retinal manifestations” needs to be changed also as many of the stated manifestations are not “retinal” such as micropthalmia, optic nerve hypoplasia or coloboma. “Ocular” would be more appropriate. In the row VAX2, the last column, “Macular dystrophy” should be changed to “macular degeneration”.
  13. Line 25, Page 2 after table 2, section CRX, Please pay attention to gene nomenclature. Firstly, all gene names should be italicised but not the protein names. A human gene is always capitalised (CRX), a mouse gene has only the first letter capitalised (Crx). Is there are reason why in line 25 there is one mouse gene (Otx1) and two human genes (OTX2 and CRX) cited? I would cite them all according to the same species whether it be human or murine.
  14. Line 37, Page 2, “to” is missing between “cone photoreceptors” and “bipolar neurons”.
  15. Line 39, Page 2, change “human Crx gene” to “human CRX gene”.
  16. Line 41, Page 2, the paper by Kawamura et al may cite CRX as a cause of autosomal dominant and recessive LCA (although this is not referenced in their text) but to my knowledge CRX is only associated with autosomal dominant LCA7. Please see Ibrahim et al Sci Rep 2018.
  17. Line 44, Page 12, cone dystrophy and macular dystrophy are two distinct clinical entities so there cannot be a cone dystrophy patient with macular dystrophy. Furthermore, in the paper by Alfano et al, VAX2 has been associated with cone dystrophy only. Please correct.
  18. Line 45, Page 16, The reference Ma et al 2017 seems inappropriately referenced here as it does not have anything to do with CRISPR/Cas in retinitis pigmentosa (title of paper cited – Monocyte infiltration and proliferation…..). I can suggest a general review article for the authors information on CRISPR/Cas in the case of IRDs, which is Sanjurjo-Soriano & Kalatzis 2018 Neural Plasticity). Furthermore, there is a general problem with reference citations throughout the manuscript. The references have been cited by name yet are not listed alphabetically but rather in order of apparition. Therefore, it is very difficult to find a reference corresponding to a citation in the text. Either list the references alphabetically or else cite them numerically in the text.
  19. Line 47, Page 16, Similarly the references Li et al and Jacobson et al are incorrect. The Li et al reference has nothing to do with LCA and the Jacobson paper concerns RPE65 gene transfer in the dog whereas the authors discuss RPE65 gene transfer to mice and monkey (I am not aware of a proof-of-concept gene transfer study in the monkey unless the authors are referring to a published toxicological study?). This section to be verified, corrected and the appropriate references included.
  20. Line 51, Page 16 to Line 52, Page 17, to my knowledge, RPE65 is not associated with AMD. It is associated with LCA type 2 and with autosomal dominant RP. Please correct the corresponding sentence.
  21. Line 11, Page 18, iPS cells were not used to treat AMD. iPSC-derived RPE was used to treat AMD. Please be careful throughout the corresponding section when speaking of “iPSC” transplantation.
  22. Line 45, page 20, the authors should be careful with the use of “multifactorial”. This can be taken to mean environmental + genetic factors thus leading to confusion. If the authors mean multiple causative genes (in keeping with the cited reference Diakatou et al), then it would be more appropriate to use the term “polygenic” or multigenic”.
  23. Line 48, page 20, I do not agree with the sentence “anterior eye sector tissue dysfunction is now recognised as the mechanism underlying a large proportion of IRDs and the cited reference Tee et al 2016 certainly does not attest to this statement either.
  24. Line 12, Page 21, I also do not agree that “iPSC-based models are not suitable for modelling retinal pathology”. Although these ex vivo systems certainly have their limitations, there are numerous examples of their being used successfully for disease modelling (Singh et al 2012, Tucker et al 2013, Cereso et al 2014, Lukovic et al 2015, Galloway et al 2017, Quinn et al 2019 just to name a few).
  25. Line 18, Page 21, IDRs should be IRDs (or should it?; see general comment above).

Author Response

Response to Reviewer 2 Comments

General comment: the authors seem to have confused inherited retinal diseases (IRDs)  with  pan-retinal  or,  in  some  cases,  pan-ocular  diseases.  IRDs  are  a  specific  group  of diseases that are characterized by loss of photoreceptors leading to, in most cases, progressive visual  loss  (although  stationary  forms  do  exist).  However,  in  this  review,  the  authors  discuss genes that are not related to IRDs, such as for example, Lmx1B, which leads to anterior segment malformations or optic neuropathy. To centre the review on IRDs, would require the removal of a large  amount  of  text,  which  seems  a  shame  when  we  consider  the  work  put  in.  Therefore,  it would be likely more appropriate to alter the title and the text to indicate that not only IRDs are treated  in  the  review.  Maybe,  for  example,  “Inherited  ocular  diseases  through  the  eyes  of homeobox genes”.

 Response: We agree that some manifestations of homeobox genes in the retina may be secondary effects because of pathological changes in other ocular tissues.

Therefore, - We corrected the title of review to “Inherited eye diseases with retinal manifestations through the eyes of homeobox genes”. In accordance with this title we made some corrections in the text:

- We changed “Inherited retinal degenerations or dystrophies (IRDs)” to “Inherited eye diseases” (page 1, lines 30-31);

- We changed the title of Table 1 “Properties of human homeobox genes associated with (IRDs)” to “Properties of human homeobox genes associated with inherited retinal diseases (IRDs) and secondary retinal malformations”;

- we changed the title of Table 2 “Cell localization and function of homeobox genes expressed in neural retina and IRDs associated” to “Cell-type specificity and function of homeobox genes expressed in neural retina and associated with inherited eye diseases in humans”.

Specific comments

Point 1: Overall,  the  review  is  comprehensible  however  there  are  grammatical  English  errors  so  I would recommend careful revision by a native English speaker. These were too numerous to cite  but  the  authors  could  keep  in  mind  that  if  a  noun  becomes  an  adjective  it  should  no longer be in plural, for example “retinal cells” as opposed to “retinal cell types”.

Response 1: We used English editing service provided by MDPI to check English of our manuscript (ID: ijms-721909) and corrected the text according their recommendations.

Point 2: Line 39, Page 1, there seems to be a typographical error. What do the authors mean by “loss-of-gain  function”?  The  term  is  usually  “loss-of-function”  or  “gain-of-function”.  Alternatively, perhaps they mean “loss- or gain-of-function”?

Response 2: page 1, line 37. "loss-of-gain function" is change to “loss- or gain-of-function”.

Point 3: Line 41, Page 1, or Line 14, Page 2, the authors use “ganglionic cells” or “ganglious cells”, respectively. Please homogenise to “ganglion cells” throughout the manuscript.

Response 3: We changed “ganglionic cells” and “ganglious cells” to “ganglion cells” throughout the manuscript (page2, line 39; page 2, line 12; page 3, line 7; page 3, line 10).

Point 4: Line 18, Page 2, “sensor neurons” should be “sensory neurons”.

Response 4: Changed “sensor neurons” to “sensory neurons” (page .2, line 26).

Point 5: Lines 34 & 35, Page 2, please change to “between bipolar and ganglion neurons”.

Response 5: “between bipolars and ganglionic neurons” was changed to “between bipolar and ganglion neurons” (page 3, lines 9-10).

Point 6: Line  46,  Page  2,  what  do  the  authors  mean  by  “microvessels  of  the  wall  cells”?  Please correct.

Response 6: “microvessels of the wall cells” was corrected on “the wall cells of the microvessels” (page 3, line 20).

Point 7: Line 1, Page 3, please change to “strict spatial and temporal regulation”.

Response 7: “strict spatial and temporalrily regulation” change to “strict spatial and temporal regulation” (page 3, lines 25-26).

Point 8: Line  24,  Page  3,  please  change  “as  differentiation  retina  proceeds”  to  “as  retinal differentiation proceeds”.

Response 8: changed “as differentiation retina proceeds” to “as retinal differentiation proceeds” (page 3, line 44).

Point 9: Line 41, Page 3, please change “conservative” to “conserved”.

Response 9: changed “conservative” to “conserved” (page 4, line 4).

Point 10: Table  1,  Please  also  define  the  homeobox  classes  in  the  foot  notes  of  the  table.  These abbreviations are defined in the text but often after the table.

Response 10: All homeobox classes were defined in the foot notes of the Table 1 (Table1, lines 54-56).

Point 11: Line  41,  Page  between  tables  1  and  2,  it  is  written  that  patients  with  ALX3  mutations  are “without significant eye manifestations”, so it is not clear why this gene is listed in table 1. However, in table 2 it is listed that it is associated with micropthalmia. Please homogenise.

Response 11: we excluded ALX3 gene from Tables 1 and 2 because of cannot find any information about its association with eye diseases in humans. Initially we found information about association this gene with microphthalmia in OMIM database but without any references to original works (see https://www.omim.org/clinicalSynopsis/136760 )

Point 12: Table 2, The title needs to be changed as IRDs are not associated in all cases. Furthermore, the title of the last column entitled “Retinal manifestations” needs to be changed also as many of the stated manifestations are not “retinal” such as micropthalmia, optic nerve hypoplasia or coloboma. “Ocular” would be more appropriate. In the row VAX2, the last column, “Macular dystrophy” should be changed to “macular degeneration”.

Response 12: Table 2,

- We changed the title of this table to “Cell-type specificity and function of homeobox genes expressed in neural retina and associated with inherited eye diseases in humans”,

- We changed the title of the last column to “Ocular manifestations in humans”,

- We changed “Macular dystrophy” to “macular degeneration” in the row VAX2, the last column.

Point 13: Line  25,  Page  2  after  table  2,  section  CRX,  Please  pay  attention  to  gene  nomenclature. Firstly,  all  gene  names  should  be  italicised  but  not  the  protein  names.  A  human  gene  is always capitalised (CRX), a mouse gene has only the first letter capitalised (Crx). Is there are reason  why  in  line  25  there  is  one  mouse  gene  (Otx1)  and  two  human  genes  (OTX2  and CRX)  cited?  I  would  cite  them  all  according  to  the  same  species  whether  it  be  human  or murine.

Response 13: We rewrote gene names in italics and capitalised human genes and proteins.

Point 14: Line 37, Page 2, “to” is missing between “cone photoreceptors” and “bipolar neurons”.

Response 14: “to” was between “cone photoreceptors” and “bipolar neurons” (page 2, line 40).

Point 15: Line 39, Page 2, change “human Crx gene” to “human CRX gene”.

Response 15: changed “human Crx gene” to “human CRX gene” (page 2, line 42).

Point 16: Line  41,  Page  2,  the  paper  by  Kawamura  et  al  may  cite  CRX  as  a  cause  of  autosomal dominant  and  recessive  LCA  (although  this  is  not  referenced  in  their  text)  but  to  my knowledge CRX is only associated with autosomal dominant LCA7. Please see Ibrahim et al Sci Rep 2018.

Response 16: Kawamura (Kawamura et al., 2010) did cite CRX association with LCA, retinitis pigmentosa and cone-rod dystrophy, but without any references (see http://mutationview.jp/MutationViewV2Server/search_top;jsessionid=E97AC9B3E8947CEEA545E461926B12D7?kind=Gene&query=crx&km=MV&commit=Go ). We brought in support of association of CRX gene with retinitis pigmentosa and cone-rod dystrophy two works (see references 110, 112).

 Point 17: Line 44, Page 12, cone dystrophy and macular dystrophy are two distinct clinical entities so there cannot be a cone dystrophy patient with macular dystrophy. Furthermore, in the paper by Alfano et al, VAX2 has been associated with cone dystrophy only. Please correct.

Response 17: We corrected text “a cone dystrophy patient with macular dystrophy and loss of outer retinal tissue” to “a cone dystrophy patient with loss of outer retinal tissue” (page 12, lines 12-13).

Point 18: Line 45, Page 16, The reference Ma et al 2017 seems inappropriately referenced here as it does not have anything to do with CRISPR/Cas in retinitis pigmentosa (title of paper cited – Monocyte  infiltration  and  proliferation…..).  I  can  suggest  a  general  review  article  for  the authors information on CRISPR/Cas in the case of IRDs, which is Sanjurjo-Soriano & Kalatzis 2018  Neural  Plasticity).  Furthermore,  there  is  a  general  problem  with  reference  citations throughout  the  manuscript.  The  references  have  been  cited  by  name  yet  are  not  listed alphabetically but rather in order of apparition. Therefore, it is very difficult to find a reference corresponding  to  a  citation  in  the  text.  Either  list  the  references  alphabetically  or  else  cite them numerically in the text.

Response 18: We cited the references numerically in order of their appearance in the text. As for wrong reference (Ma et al., 2017), we exclude it and changed to Sanjurjo-Soriano, Kalatzis, 2018 (as you recommend) and Diakatou et al., 2019 (Genome Editing as a Treatment for the Most Prevalent Causative Genes of Autosomal Dominant Retinitis Pigmentosa) (references 7 and 340) (page 15, line 49).

Point 19: Line 47, Page 16, Similarly the references Li et al and Jacobson et al are incorrect. The Li et al  reference  has  nothing  to  do  with  LCA  and  the  Jacobson  paper  concerns  RPE65  gene transfer in the dog whereas the authors discuss RPE65 gene transfer to mice and monkey (I am not aware of a proof-of-concept gene transfer study in the monkey unless the authors are referring to a published toxicological study?). This section to be verified, corrected and the appropriate references included.

Response 19: We excluded wrong references (Li et al., 2011, and Jacobson et al., 2006) and changed them to Bennicelli et al., 2008 (ref. [341], page 15, line 51) which concerning mouse model of LCA. We also removed reference to monkeys in the text (page 15, line 50).

Point 20: Line 51, Page 16 to Line 52, Page 17, to my knowledge, RPE65 is not associated with AMD. It  is  associated  with  LCA  type  2  and  with  autosomal  dominant  RP.  Please  correct  the corresponding sentence.

Response 20: we removed AMD from the text (page 16, line 2).

Point 21: Line 11, Page 18, iPS cells were not used to treat AMD. iPSC-derived RPE was used to treat AMD.  Please  be  careful  throughout  the  corresponding  section  when  speaking  of  “iPSC” transplantation.

Response 21: changed “iPS cells” to “iPSC-derived RPE cells” (page 17, line 10).

Point 22: Line 45, page 20, the authors should be careful with the use of “multifactorial”. This can be taken to mean environmental + genetic factors thus leading to confusion. If the authors mean multiple causative genes (in keeping with the cited reference Diakatou et al), then it would be more appropriate to use the term “polygenic” or multigenic”.

Response 22: “multifactorial” was changed to “multigenic” (page 4 before table 1, line 25; and page 19, line 37).

Point 23: Line 48, page 20, I do not agree with the sentence “anterior eye sector tissue dysfunction is now  recognised  as  the  mechanism  underlying  a  large  proportion  of  IRDs  and  the  cited reference Tee et al 2016 certainly does not attest to this statement either.

Response 23: We removed the sentence “As such, anterior eye sector tissues dysfunction is now recognized as the mechanism underlying a large proportion of IRDs (Tee et al., 2016)” (page 19, lines 39-40). The development of secondary eye pathologies is described in other part of our review (see page 4, lines 27-37).

Point 24: Line 12, Page 21, I also do not agree that “iPSC-based models are not suitable for modelling retinal pathology”. Although these ex vivo systems certainly have their limitations, there are numerous examples of their being used successfully for disease modelling (Singh et al 2012, Tucker et al 2013, Cereso et al 2014, Lukovic et al 2015, Galloway et al  2017,  Quinn  et  al 2019 just to name a few).

Response 24: change “but nonetheless are not suitable for modelling retinal pathology in some cases” to more careful statement “for modelling retinal pathologies, but certainly have some limitations” (page 20, lines 3-5), and cited some references in support (ref. 391-393).

Point 25: 25. Line 18, Page 21, IDRs should be IRDs (or should it?; see general comment above).

Response 25: We agree with you that some inherited diseases associated with homeobox genes can primarily develop not in the retina but in other ocular tissues.

So, we changed IDRs to “inherited eye diseases in general, and IRDs in particular” (page 20, lines 12-13).

Round 2

Reviewer 1 Report

Thanks for the revision. I have no further comments.